# Mechanism of Na⁺-dependent citrate transport from the structure of an asymmetrical CitS dimer

David Wöhlert, Maria J Grötzinger, Werner Kühlbrandt*, Özkan Yildiz*

Department of Structural Biology, Max Planck Institute of Biophysics, Frankfurt am Main, Germany

**Abstract** The common human pathogen *Salmonella enterica* takes up citrate as a nutrient via the sodium symporter SeCitS. Uniquely, our 2.5 Å x-ray structure of the SeCitS dimer shows three different conformations of the active protomer. One protomer is in the outside-facing state. Two are in different inside-facing states. All three states resolve the substrates in their respective binding environments. Together with comprehensive functional studies on reconstituted proteoliposomes, the structures explain the transport mechanism in detail. Our results indicate a six-step process, with a rigid-body 31° rotation of a helix bundle that translocates the bound substrates by 16 Å across the membrane. Similar transport mechanisms may apply to a wide variety of related and unrelated secondary transporters, including important drug targets.

## Introduction

Citrate transporters are found in a wide range of bacteria, archaea and eukaryotes. Bacteria use specific transporters (*Sobczak and Lolkema, 2005*) to take up di- and tricarboxylates as a carbon source (*Mulligan et al., 2014*; *Pos et al., 1998*). The human citrate transporter NaCT plays a central role in fatty acid synthesis and glycolysis (*Gopal et al., 2007*), and is a potential drug target against obesity and diabetes (*Liang et al., 2015*). The *Drosophila* INDY gene encodes a related dicarboxylate transporter implicated in fat storage (*Rogina et al., 2000*). The x-ray structure of VcINDY, a homologous dicarboxylate transporter from *Vibrio cholerae* is known in the inward-facing state (*Mancusso et al., 2012*). Unexpectedly, a recent cryo-EM structure of the citrate transporter KpCitS from *Klebsiella pneumoniae* (*Kebbel et al., 2013*) revealed a similar overall domain architecture to VcINDY (*Mancusso et al., 2012*) and to archaeal Na⁺/H⁺ antiporters of the NhaP family (*Goswami et al., 2011*; *Paulino et al., 2014*; *Wöhlert et al., 2014*), in both cases without detectable sequence homology. CitS, VcINDY and the NhaP antiporters all form homodimers of two protomers, each organized in a helix bundle and a dimer contact domain (*Kebbel et al., 2013*; *Mancusso et al., 2012*; *Vinothkumar et al., 2005*), which suggests similar transport mechanisms.

## Results and discussion

In membrane vesicles (*Lolkema et al., 1994*; *van der Rest et al., 1992*) and proteoliposomes (*Pos and Dimroth, 1996*), CitS from *Klebsiella pneumoniae* (KpCitS) was previously shown to transport citrate as HCit²⁻ in a sodium-dependent manner. We observed similar transport properties for CitS from *Salmonella enterica* (SeCitS), which is closely related to KpCitS. The two homologues share a remarkably high sequence identity of 92% (*Figure 1*), indicating that their transport mechanisms must be very similar. Iso-citrate and, to a lesser extent, malate inhibit Na⁺-dependent ¹⁴C-citrate uptake by SeCitS into proteoliposomes. Succinate, α-ketoglutarate, and glutaric acid reduce uptake slightly, whereas tricarballylic acid, which lacks the citrate hydroxyl group, has no effect (*Figure 2A*).

*For correspondence: werner. kuehlbrandt@biophys.mpg.de (WK); Oezkan.Yildiz@biophys. mpg.de (ÖY)

**eLife digest** Cells have specialized proteins known as transporters in their surface membranes that move molecules into or out of the cell. Transporters pass their cargo through the membrane by changing shape. This process requires energy and is sometimes driven by simultaneously transporting a charged ion such as sodium. There are different classes of transporters and researchers have described a range of structural changes, and therefore transport mechanisms, that different transporters use.

Citrate transporters are found in a wide range of organisms. In bacteria, they bring the citrate substrate molecule into the cell to be used as a nutrient. In humans, citrate transporters are important in metabolism, and are of interest as targets for drugs that could potentially treat obesity and diabetes. This requires an understanding of the atomic structure and the transport mechanisms used by citrate transporters, which were not known.

Wöhlert et al. now use a technique called X-ray crystallography to uncover the structure of a citrate transporter called SeCitS in high detail. This transporter is found in a bacterium called *Salmonella enterica*, a well-known human pathogen that causes typhoid. The crystallized protein simultaneously showed three different structural states – one where the citrate binding site faces the outside of the cell, and two where the binding site faces the inside of the cell. The simultaneous occurrence of different functional states in one and the same crystal structure of a membrane transporter is so far unique.

Combining the detailed structures of SeCitS with biochemical studies allowed Wöhlert et al. to deduce that citrate is transported in a six-step process. Sodium ions attach to SeCitS, and then citrate binds to the transporter from outside the cell. This binding causes part of the protein to undergo a substantial rotation, shifting it to an inward-facing state and moving the citrate and sodium ions inside the cell. The release of the citrate and sodium ions then triggers the reverse rotation of the transporter, bringing the empty binding site back to the outside of the cell for a repeat of the cycle.

After working out the mechanisms of a bacterial citrate transporter, the next challenge is to extend the analysis to the structure of similar transporters in more complex organisms, including human cells. This could provide an accurate basis for drug development.

This demonstrates the specificity of the CitS binding site for 2-hydroxycarboxylates. Malate, which is smaller than citrate, inhibits citrate uptake by SeCitS but is not transported (*Figure 2B*). Citrate symport is driven by $Na^+$ but not by $K^+$ or $Li^+$ (*Figure 2C, D*), demonstrating the exquisite specificity of SeCitS for $Na^+$ ions. Sodium transport is cooperative with a Hill coefficient of 1.89, whereas citrate is not, suggesting that citrate transport is coupled to at least two $Na^+$ ions (*Figure 3*). SeCitS is active between pH5 and pH8 with an optimum at pH7, resulting in a roughly bell-shaped pH profile (*Figure 3—figure supplement 1*). Down-regulation of transport at low pH can be attributed to a limitation in sodium binding, while at elevated pH the availability of the $HCit^{2-}$ citrate species is limiting. Citrate uptake is enhanced at lower outside pH (*Figure 3—figure supplement 2A*). Under these conditions, transport by SeCitS is electroneutral, since valinomycin has no effect (*Figure 3—figure supplement 2B*). A lower outside pH would shift the citrate buffer equilibrium towards $HCit^{2-}$. Therefore, a low outside pH increases the local substrate concentration, while a high inside pH tends to deprotonate the $HCit^{2-}$ substrate and thus removes it from the transport equilibrium. We conclude that protons do not participate directly in the transport mechanism. This conclusion is substantiated by the observation that an increase in the internal $Na^+$ concentration does not stimulate citrate uptake (*Figure 3—figure supplement 2C*), which argues against a previously proposed citrate/proton symport (or citrate/hydroxide antiport) in exchange for internal sodium (*Pos and Dimroth, 1996*).

To understand the mechanism in detail we determined the structure of CitS from the human pathogen *Salmonella enterica* (*Figure 4*) by single-wavelength anomalous dispersion with crystals of seleno-methionine derivatized protein (*Figure 4—figure supplement 1*). Phases were extended to the 2.5 Å diffraction limit of native crystals (*Table 1*). The asymmetric unit contains two homodimers of two protomers in different conformations (*Figure 4A, B*). Each protomer has 13 helix elements

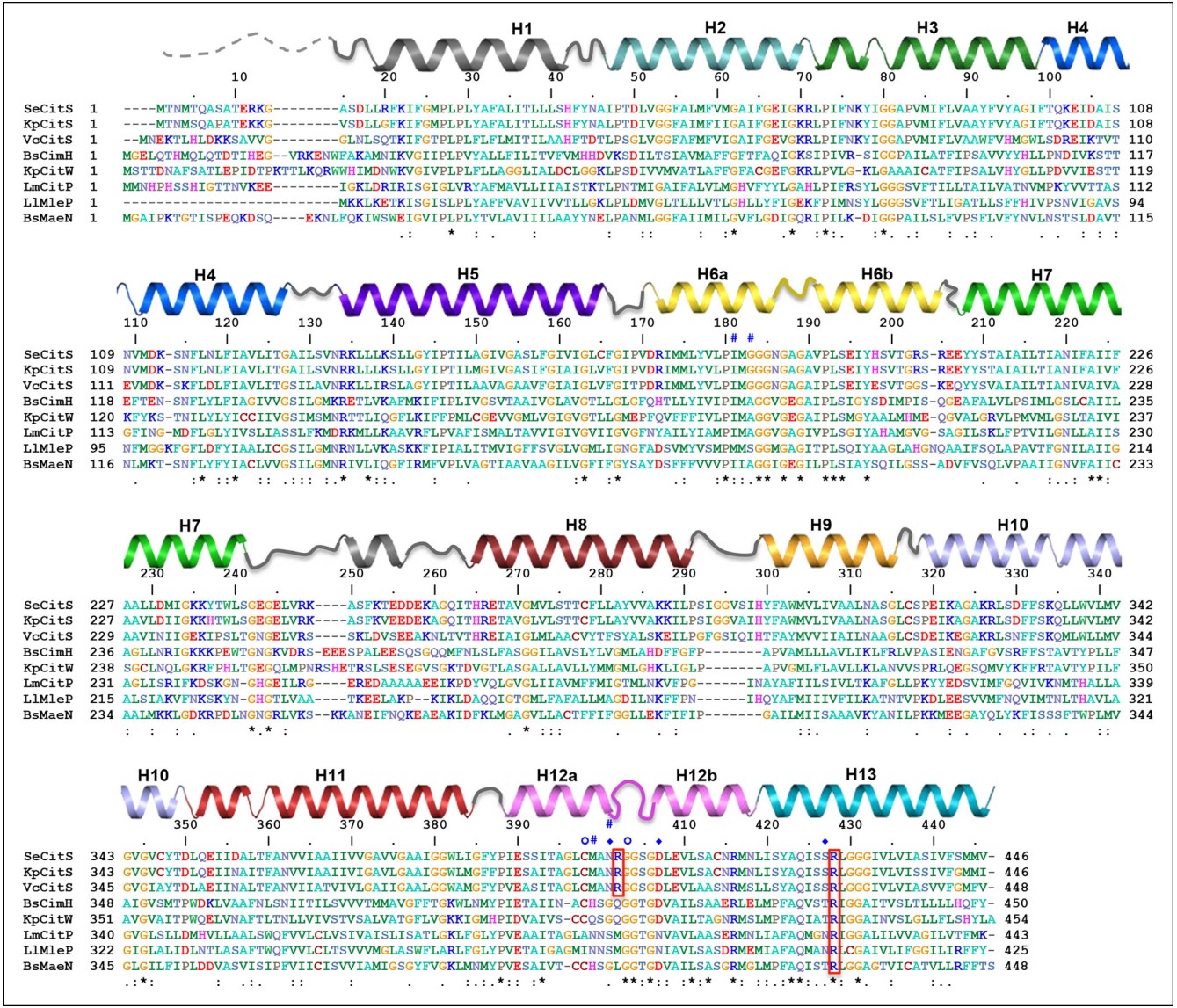

**Figure 1.** Sequence alignment. Sequence alignment of 2-hydroxycarboxylate transporters. The secondary structure of SeCitS is shown above the alignment. R402 and R428 of the citrate-binding site are outlined in red. Symbols above the sequence indicate residues involved in sodium binding. A hashtag (#) marks the residues that form the Na1 site. Residues with sidechains coordinating Na2 are marked with a diamond (♦), and those that coordinate Na2 with backbone carbonyls with an open circle (O). Most of the conserved residues (*) are found in the two helix hairpins H6 and H12, and in transmembrane helix H13.

SeCitS: Citrate/sodium symporter from *Salmonella Enterica* (WP_024797394.1)
KpCitS: Citrate/sodium symporter from *Klebsiella pneumoniae* (WP_025860623.1)
VcCitS: Citrate/sodium symporter from *Vibrio_cholerae* (WP_001003397.1)
BsCimH: Citrate/malate transporter from *Bacillus_subtillis*, (P94363.1)
KpCitW: Citrate/acetate transporter from *Klebsiella_pneumoniae*, (AF411142.1)
LmCitP: Citrate transporter from *Leuconostoc_mesenteroides* (AAA60396.1)
LlMleP: Malate transporter from *Lactococcus lactis*, (CAA53590.1)
BsMaeN: Malate/sodium symporter from *Bacillus_subtilis*, (AFQ59004.1)

(H1–H13), including eleven transmembrane helices (TMH) and two helix hairpins (H6, H12), with the N-terminus on the cytoplasmic side and the C-terminus on the outside. Helices H2-–7 and H8–13 are organized in two repeats with inverted topology (*Figure 4C*), connected by a flexible cytoplasmic loop (*Figure 4B*). Together, helices H5–7 of repeat 1 and H11–13 of repeat 2 form a bundle on either side of the central contact domains, which hold the dimer together through extensive

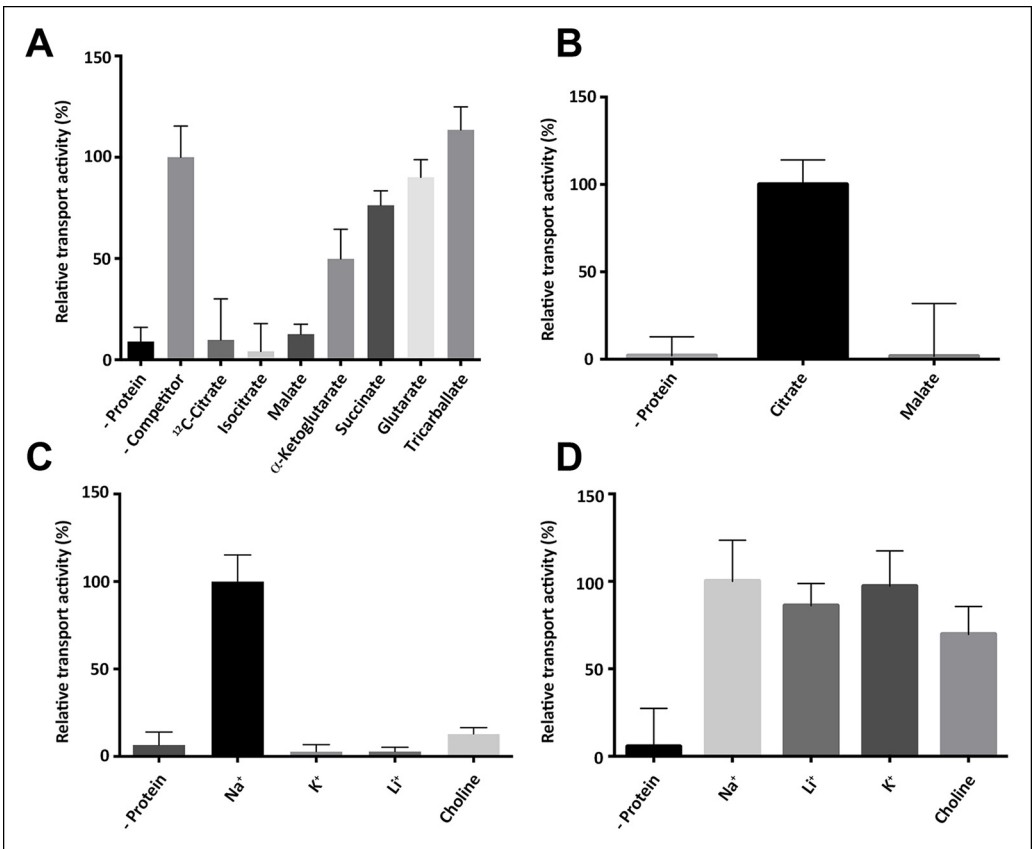

**Figure 2.** Substrate specificity of SeCitS. (**A**) The substrate specificity of SeCitS was established by a proteoliposome uptake inhibition assay. Potential substrates or competitors were added in thousandfold excess of $^{14}$C-citrate (5 µM) and transport was measured. The 2-hydroxycarboxylates malate and iso-citrate inhibit $^{14}$C-citrate uptake completely. α-Ketoglutarate, which has a carbonyl instead of the citrate hydroxyl group, inhibits less strongly. Succinate and glutarate inhibit transport only slightly. Tricarballate has no effect. (**B**) While malate inhibits citrate uptake, it is not a substrate for SeCitS, as uptake of $^{14}$C-malate (43 µM) is not detectable. (**C,D**) SeCitS is highly specific for Na$^+$. Neither Li$^+$ nor K$^+$ drive (**C**) or inhibit (**D**) citrate uptake. Choline was used as a negative control in both assays. Initial uptake rates were plotted relative to (**A**) absence of competitor, (**B**) citrate transport or (**C,D**) sodium-driven transport.

hydrophobic interactions of H2, 4, 8 and 10. A 16 Å-deep hydrophobic cavity on the cytoplasmic side of the dimer interface contains the hydrophobic tail of a detergent or lipid molecule (*Figure 5A*).

The two dimers in the asymmetric unit are similar, with an overall rmsd of 0.5 Å, whereas the protomers within one dimer differ substantially by an rmsd of 8.4 Å. The most conspicuous differences are manifest in the vertical positions of the two hairpins H6 and H12 in the helix bundle (*Figure 5A*). Comparing the two protomers of dimer 1, the C-terminal end of H6 projects 16 Å above the outer membrane surface in protomer A, while it hardly protrudes in protomer B. Conversely, the cytoplasmic H12 ends roughly at the inner membrane surface in protomer A, but extends 13 Å above it in protomer B. The relative position of helices and hairpins within each bundle is unchanged. Evidently, the whole bundle moves as a rigid body from its position in protomer A to that in protomer B, while the central dimer contact domain remains static. The crystal contacts of both dimers in the asymmetric unit are different. Since the polyptide structures of the two dimers are almost identical, the observed asymmetry cannot be attributed to crystal packing. Dimer asymmetry is equally striking with respect to surface structure and electrostatic potential distribution. Protomer A has more positive charges on the periplasmic side than protomer B (*Figure 6A*). On the cytoplasmic side, positive charges predominate on the surface of protomer B, while positive and negative charges are roughly evenly distributed on protomer A (*Figure 6B*). Overall, positive charges dominate on the cytoplasmic

side of the dimer (*Figure 6C, 6D*), in line with the positive-inside rule for membrane proteins (*Nilsson and von Heijne, 1990*; *von Heijne, 1992*).

Short stretches of glycine-rich unwound polypeptide link the two halves of hairpins H6 and H12. Together they define the substrate-binding site (*Figure 5C–E*) at the interface between the helix bundle and the dimer contact domain. On the extracellular surface, the binding site is found in a ~6 Å deep cavity of protomer A (*Figure 6A*), while in protomer B it is located at the bottom of a ~13 Å-deep channel on the cytoplasmic side (*Figure 6B*). We conclude that protomer A is outward-facing and that protomer B faces inward. The binding sites in both protomers are strongly positively charged (*Figure 6A, B*). Two detergents and one lipid molecule were identified on the periphery of the dimer. A further detergent molecule was situated in a hydrophobic cavity between the central dimer contact domain and the six-helix bundle of the outward-facing protomer A (*Figure 6C–E*). H5 in this bundle is straight in protomer A but kinked near its cytoplasmic end in protomer B, to accommodate the movement of the helix bundle (*Figure 6E, F*).

All four protomers show clear electron density for citrate in the binding site (*Figure 7*). In the outward-facing protomers, the citrate is closely coordinated by two arginines (Arg402, Arg428), two polar sidechains (Asn186, Ser405) and the protein backbone of both hairpins (*Figure 5 and 7A*). The only residue in the static contact domain involved in substrate coordination is Tyr348 in H10, which forms a $\pi$-$\pi$-interaction with a citrate carboxyl. One ordered water molecule participates directly in citrate binding. Its trigonal-bipyramidal coordination geometry (*Figure 7*) might suggest a $Na^+$ ion rather than water, which would imply that the transported entity is $NaCit^{2-}$ rather than $HCit^{2-}$. Because the electron density is weak and the coordination distance of >2.8 Å is longer than would be expected for $Na^+$, we interpret this density as a water molecule. In both outward-facing protomers, two $Na^+$ ions are clearly resolved next to the citrate (*Figures 5C and 7A*). In the Na1 site, four backbone carbonyls in the unwound hairpin stretches coordinate one $Na^+$. In the Na2 site, the carboxyl group of Asp407, the polar sidechains of Asn401, Ser427 and the backbone carbonyls of Cys398 and Gly403 coordinate the ion.

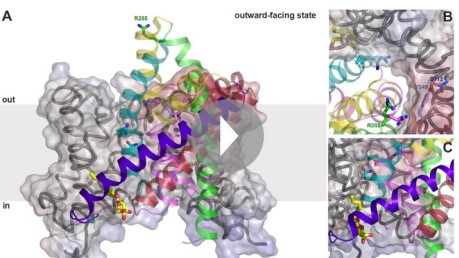

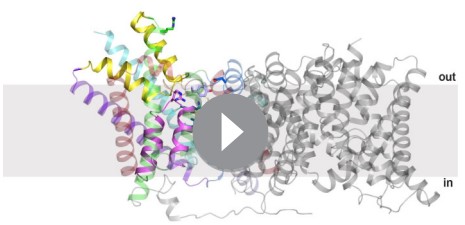

**Video 1.** Schematic representation of SeCitS transport. The movie shows a morph from the outward-facing to the inward-facing conformation for one protomer of the SeCitS dimer. Arg402, Arg428 and Tyr348, which coordinate citrate in the outward-facing conformation, are drawn as stick models, while the $Na^+$ ions are represented as grey spheres. $Na^+$ ions bind to their respective sites in the helix bundle, followed by citrate binding between helix bundle and dimer contact domain. Subsequently, the substrates are translocated by a rotation of the bundle. Citrate release is independent from the release of either $Na^+$ ion. Due to the empty Na2 binding site in protomer B' we assume that this ion is released immediately after the citrate. After substrate release the empty transporter changes its conformation back to the outward-facing state to repeat the cycle.

**Video 2.** Schematic representation of domain, helix and sidechain movements. Three synchronized movies show different views of one SeCitS protomer during the transport cycle: (**A**) from the membrane plane, (**B**) in the perpendicular direction from the cell exterior and (**C**) a detailed view of the substrate-binding site and the detergent/lipid binding pocket. Helices of the rotating bundle domain are coloured, while helices in the static dimer contact domain are shown in grey behind their corresponding transparent electrostatic surface. The negatively charged periplasmic surface of SeCitS (transparent red) attracts Arg205 of H7 (green), which, in the inward-facing state, forms an ion bridge to Asp112 in H4 and a hydrogen bond to Tyr348 (lavender) in H10 of the dimer contact domain. In the outward-facing state, Asp112 interacts with Tyr348, which rotates to block access to the substrate-binding site in the inward-facing state. A detergent molecule (yellow) in the hydrophobic pocket between H5 (purple), H13 (cyan) and the dimer contact domain, is displaced in the inward-facing state by the movement of H13. H5, which is straight in the outward-facing state, kinks during the bundle rotation to prevent its partial exposure to the cytoplasm.

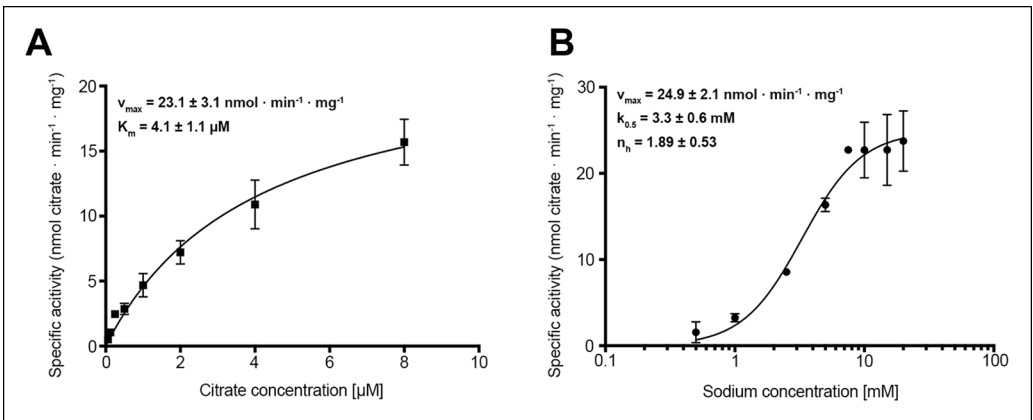

**Figure 3.** Citrate and sodium transport kinetics. (**A**) Citrate uptake by SeCitS containing proteoliposomes in presence of 25 mM Na$^+$ is non-cooperative and follows Michealis-Menten kinetics with a $K_m$ of 4.1 µM and a $v_{max}$ of 23.1 nmol · min$^{-1}$ · mg$^{-1}$. (**B**) Na$^+$ transport in presence of 5 µM citrate is cooperative, with a Hill coefficient of 1.89. The affinity of SeCitS for Na$^+$ is lower than for citrate, as demonstrated by a $K_m$ of 3.3 mM. The $v_{max}$ of 24.9 nmol · min$^{-1}$ · mg$^{-1}$ indicates a turnover rate of 1.2 citrate molecules per protomer per minute at room temperature.

The following figure supplements are available for Figure 3:

**Figure supplement 1.** pH-dependence of SeCitS transport.

**Figure supplement 2.** Driving force, electrogenicity and effect of internal salt concentration.

Two ordered water molecules participate in Na$^+$binding, one of them suspended between the two Na$^+$ ions (***Figures 5C and 7A***), accounting for the observed cooperativity of Na$^+$ transport (***Figure 3B***). Asn401, which coordinates Na1 with its backbone carbonyl and Na2 via its side chain, may contribute to this effect.

There is no difference in substrate coordination or in main-chain conformation between the two outward-facing protomers A and A' (rmsd 0.5 Å). A and A' can therefore be considered as identical. Interestingly, the main chain conformations of the two inward-facing protomers B and B' are likewise practically identical (rmsd 0.6 Å), but the citrate and Na$^+$ coordination in B and B' is clearly different. In protomer B, citrate is partially hydrated and coordinated by the hydroxyl of Ser405 and the backbone carbonyl of Gly404 in the conserved GGXG motif of H12 (***Figures 1, 5D and 7B***). Both Na$^+$ sites are occupied and take up the same position relative to the citrate as in the outward-facing state. In protomer B', the Na2 site is empty, even though the structure of the ion-coordinating hairpin hardly changes (***Figures 5E and 7C***). The citrate is fully hydrated and not directly attached to a sidechain, and a second citrate is present near Gln424 in H7.

The rigid-body movement of the helix bundle from its position in protomers A and A' to that in protomers B or B' can be described as a 31° arc-like rotation around an axis roughly parallel to the membrane and perpendicular to the long dimer axis (***Figures 5, 7*** and ***Video 1***). The rotation is facilitated by the greasy interface between the helix bundle and the static dimer contact domain. The greasy interface consists almost entirely of small hydrophobic sidechains and a bound detergent molecule that may take the place of a membrane lipid alkyl chain (***Figures 5 and 8 A-D***). During the bundle rotation the detergent molecule is displaced by H13. As the helix bundle reaches the inward-facing position, the straight, hydrophobic helix H5 kinks at Gly143, thus preventing its partial exposure to the cytoplasm, and an ion bridge forms between Asp112 and Arg205 in H7 (***Figure 8E, F***; ***Video 2***). As a result of the helix bundle rotation, the binding site with the bound citrate moves by 16 Å from the external membrane surface in the outward-facing state to a position where it is accessible from the cytoplasmic membrane surface in the inward-facing state (***Figure 5***). Since transport is non-cooperative with respect to citrate (***Figure 3A***), we conclude that the two binding sites in the dimer act independently of one another.

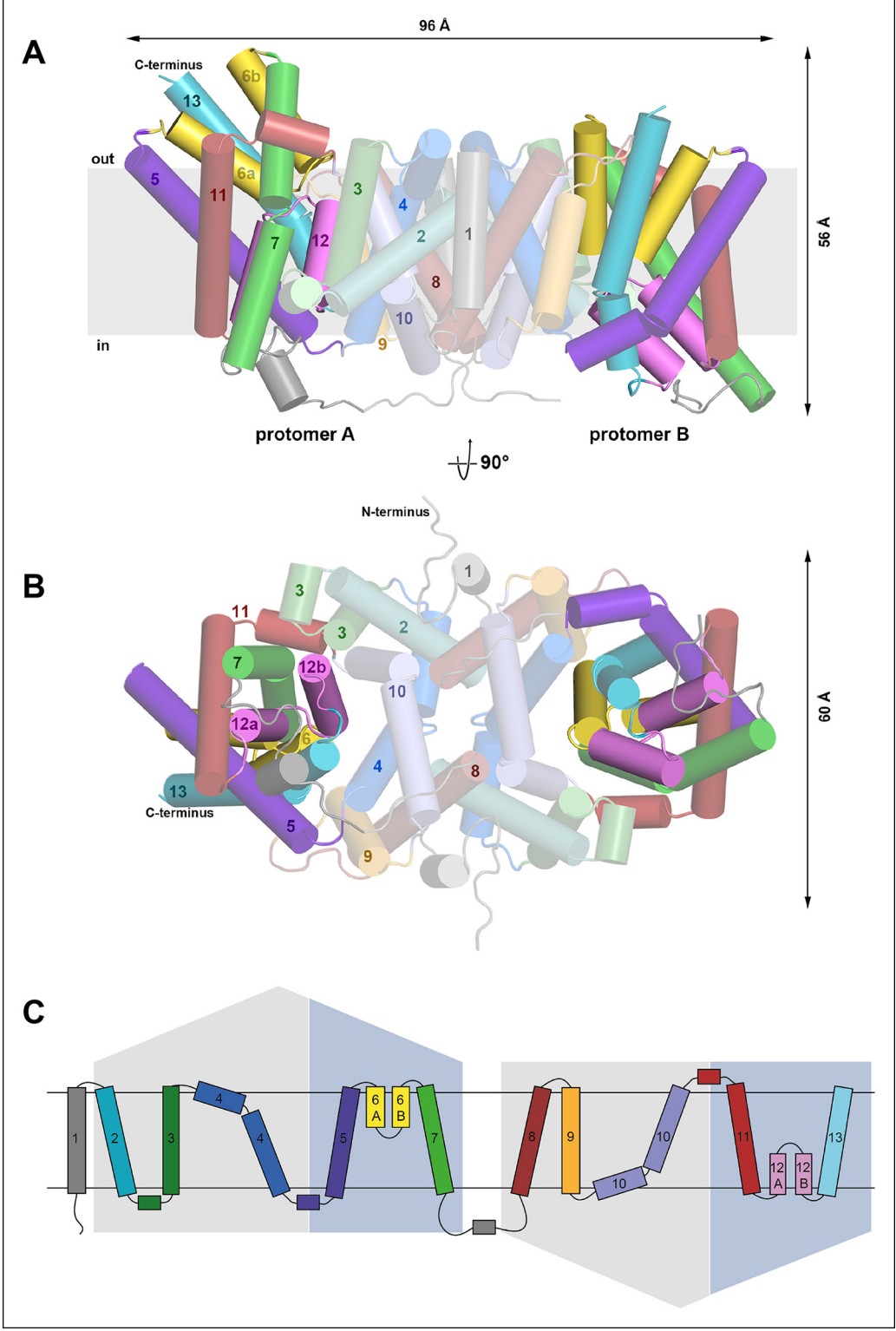

**Figure 4.** Overall structure of SeCitS and topology. Side view (**A**) and cytoplasmic view (**B**) of the SeCitS homodimer. The dimer is oval, with a long axis of 96 Å and a short axis of 60 Å. Each protomer consists of eleven transmembrane helices and two helix hairpins (yellow and pink). (**C**) SeCitS consists of two inverted 5-TMH repeats connected by a long cytoplasmic loop plus an additional N-terminal helix. Each repeat contains one hairpin. Helices belonging to the helix bundle are shown on blue background, while helices of the dimer contact domain

*Figure 4. continued on next page*

*Figure 4. Continued*

are shown on grey background. The extended flexible link between the two inverted repeats is completely resolved in protomer A (**A**).

The following figure supplements are available for Figure 4:

**Figure supplement 1.** SeMet phasing.

Unlike the $Na^+$ ions, the citrate di-anion is not occluded by the hairpin loops in SeCitS. Similarly, the dicarboxylate substrate is not occluded in VcINDY (*Mancusso et al., 2012*), whereas the corresponding substrate is occluded within the helix bundle of GltPh (*Boudker et al., 2007*). SeCitS may lack a well-defined substrate-occluded state, but the citrate would effectively be occluded during the transition from the outward-facing to the inward-facing state, while the occupied binding site rotates past the hydrophobic surface of the dimer interface domain (*Video 2*).

In the outward-facing state, strong polar and ionic interactions facilitate citrate binding at low ambient substrate concentrations (*Figure 3A and 5C*). In the inward-facing state, the binding affinity for the substrate is reduced (*Pos and Dimroth, 1996*), so the citrate can detach. We propose that the three citrate positions we observe in the two inward-facing protomers mark the path of the substrate during its release from the binding site past the highly conserved Arg428 (*Figures 1, 5 and 7*), along a trajectory that guides the negatively charged substrate towards the cytoplasm, where it is metabolized. Partial release of the citrate di-anion would weaken $Na^+$ binding, which explains why only one Na site is occupied in B'. Since transport is electroneutral, both $Na^+$ ions must dissociate from the inward-facing state. MD simulations suggest that in other Na-dependent transporters such as LeuT (*Grouleff et al., 2015*), GltPh (*Zomot and Bahar, 2013*), vSGLT (*Watanabe et al., 2010*), at least one of the $Na^+$ ions is released before the main substrate. In the case of SeCitS, comparison of the inward-facing protomers B and B' indicates unambiguously that citrate is released before $Na^+$, and that Na2 is released before Na1 (*Figure 5D, E*). Once the citrate has left the binding site, the helix hairpins or H13 would need to rearrange to release Na1, while a minor reorientation of the Asp407 or Ser427 sidechains is sufficient to release Na2.

Comparison of the binding sites in the outward-facing protomers indicates that both $Na^+$ ions have to be in place before citrate can bind. A cryo-EM structure from 2D crystals of the closely related KpCitS from *Klebsiella pneumoniae* found that sodium citrate induced a major conformational change in the helix bundle, whereas potassium citrate did not (*Kebbel et al., 2013*), supporting the proposed binding order. Therefore the complete transport mechanism entails the following six steps: (1) The Na sites are occupied by $Na^+$ in the outward-facing state; (2) a citrate binds from the external medium; (3) citrate binding triggers the arc-like rotation of the helix bundle in the transition from the outward-facing to the inward-facing state; (4) in the inward-facing state, the citrate becomes hydrated and diffuses into the cytoplasm; (5) the sodium ions come off; (6) the release of all substrates enables the reverse arc-like rotation of the helix bundle to expose the empty binding site again to the cell exterior, and the cycle repeats (*Figure 9*; *Videos 1 and 2*).

Notwithstanding the large domain movements associated with substrate translocation, citrate exchange rates are high, with a turnover of up to 137 $s^{-1}$ reported for the closely related KpCitS (*Pos and Dimroth, 1996*). Citrate uptake by SeCitS is substantially slower at 1.2 molecules per minute (*Figure 3*). Therefore, the arc-like rotation of the helix bundle that translocates the bound substrate across the membrane is not rate-limiting. The same seems to hold true for GltPh, which shows a slow substrate uptake rate of 0.29 molecules per minute (*Ryan et al., 2009*), while crosslinking experiments show that the conformational change happens within seconds (*Reyes et al., 2009*). Assuming that the $Na^+$ concentration does not limit substrate binding or release under physiological conditions, the rate-limiting step in SeCitS is most likely the reverse rotation of the helix bundle with the binding site empty.

An influence of lipids on the conformational dynamics in GltPh by inserting a lipid molecule between both domains was recently proposed by MD simulations (*Akyuz et al., 2015*). The structure of SeCitS offers experimental evidence for the existence of a hydrophobic pocket at the interface of both domains and highlights the importance of the bilayer for the activity of membrane transporters.

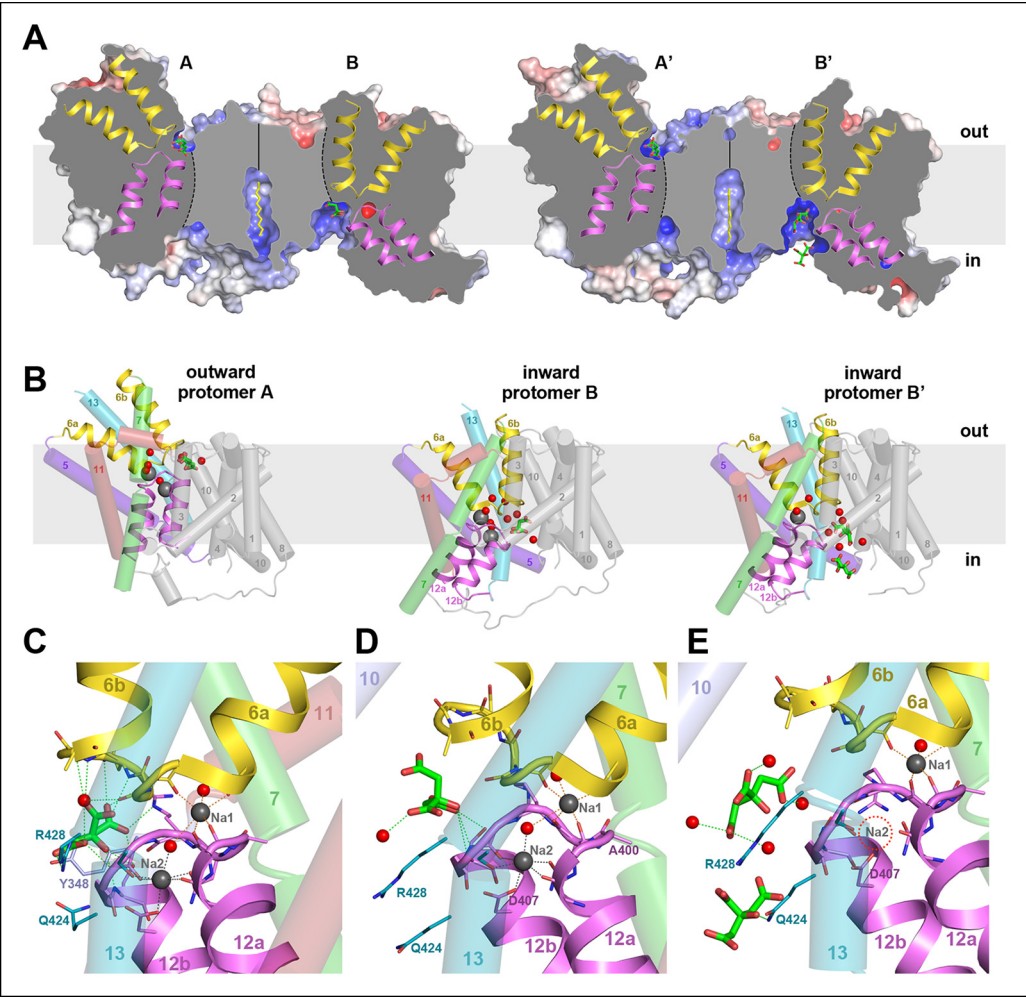

**Figure 5.** Two different states of the asymmetrical SeCitS dimer. (**A**) The outward-facing protomers A and A' bind citrate in a shallow, positively charged cavity between the helix bundle and dimer contact domain. In the inward-facing protomers B and B', citrate binds in a deep cytoplasmic cavity. In B', two citrate molecules are resolved. (**B**) In protomers A, A' and B, two Na$^+$ are occluded in the helix bundle, while in B' only one Na$^+$ is present. The substrates are translocated 16 Å across the membrane by a 31° rotation of the helix bundle relative to the static dimer contact domain. (**C**) In the outward-facing protomers, citrate is closely coordinated by sidechains of both hairpins and H13. Neither Na$^+$ participates directly in citrate coordination. (**D**) In the inward-facing protomer B, citrate is hydrated and attached weakly to the glycine-rich loop of H12. The Na1 and Na2 sites in (**C**) and (**D**) are virtually identical, indicating that the transition from the outward-facing to the inward-facing state does not affect Na$^+$-coordination geometry. (**E**) In protomer B', only the Na1 site is occupied. Two citrate molecules are resolved, outlining a likely trajectory for citrate release (**Video 1**).

It remains to be seen whether any of the lipid-binding sites in these transporters are structurally conserved.

The arc-like rotation of the helix hairpins in SeCitS is reminiscent of the recently proposed conformational change for substrate translocation in GltPh (*Crisman et al., 2009*; *Reyes et al., 2009*; *Verdon et al., 2014*). In the Na$^+$/H$^+$ antiporters (*Lee et al., 2013*; *Paulino et al., 2014*) or the bile acid transporter ASBT (*Zhou et al., 2014*), where the binding site is defined by unwound stretches of two trans-membrane helices in a structurally homologous bundle, this process also involves a rotation of the bundle around a similar axis as in SeCitS, although the movement is significantly smaller. The domain structure of the unrelated transporter YdaH (*Bolla et al., 2015*) bears a striking resemblance to that of SeCitS, suggesting that it may work in the same way. The rotating arc mechanism

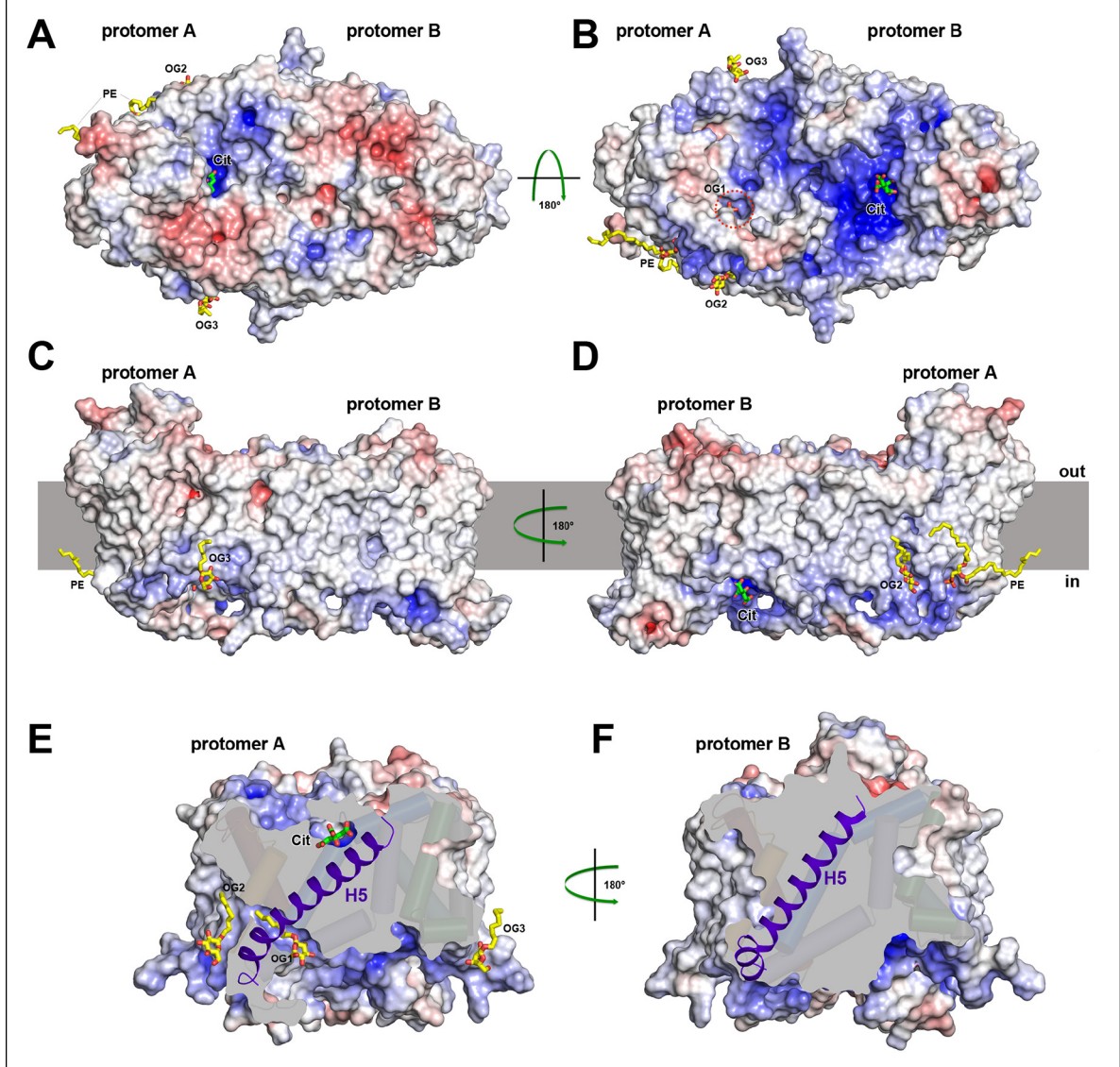

**Figure 6.** Electrostatic surface potential and bound detergent/lipid molecules. Exterior (**A**) and cytoplasmic views (**B**) of the electrostatic surface potential of SeCitS accentuates the dimer asymmetry. The binding sites for the citrate di-anion (green) on the exterior surface of protomer A and the cytoplasmic side of protomer B are strongly positively charged (dark blue). (**C, D**) Positions of bound detergent and lipid molecules (yellow) are shown in the side view of the electrostatic surface. Apart from the aliphatic chain in the hydrophobic cavity of the dimer interface (**Figure 5**), they are positioned close to the helix bundle. (**E**) In the outward-facing protomers, a hydrophobic cavity between H5, H13 and the dimer contact domain is filled by a detergent molecule. This cavity is closed in the inward-facing protomers (**F**).

described here for SeCitS thus seems to apply to a large class of secondary membrane transporters with unwound helix elements or hairpins that were previously thought to be unrelated.

# Materials and methods

## Protein expression and purification

A gene coding for CitS from *Salmonella enterica* (WP_000183608) was cloned into a pET21d plasmid harboring an N-terminal His$_{10}$-Tag and a thrombin cleavage site between tag and target protein. The resulting plasmid was used to transform *E. coli* C41-(DE3) cells. After expression for 10 h at 37°C in ZYM-5052 autoinduction medium (*Studier, 2005*) cells were harvested, resuspended in 20 mM Tris/HCl pH7.4, 150 mM NaCl, 5 mM EDTA, 5 mM β-mercaptoethanol (β-ME) and broken using

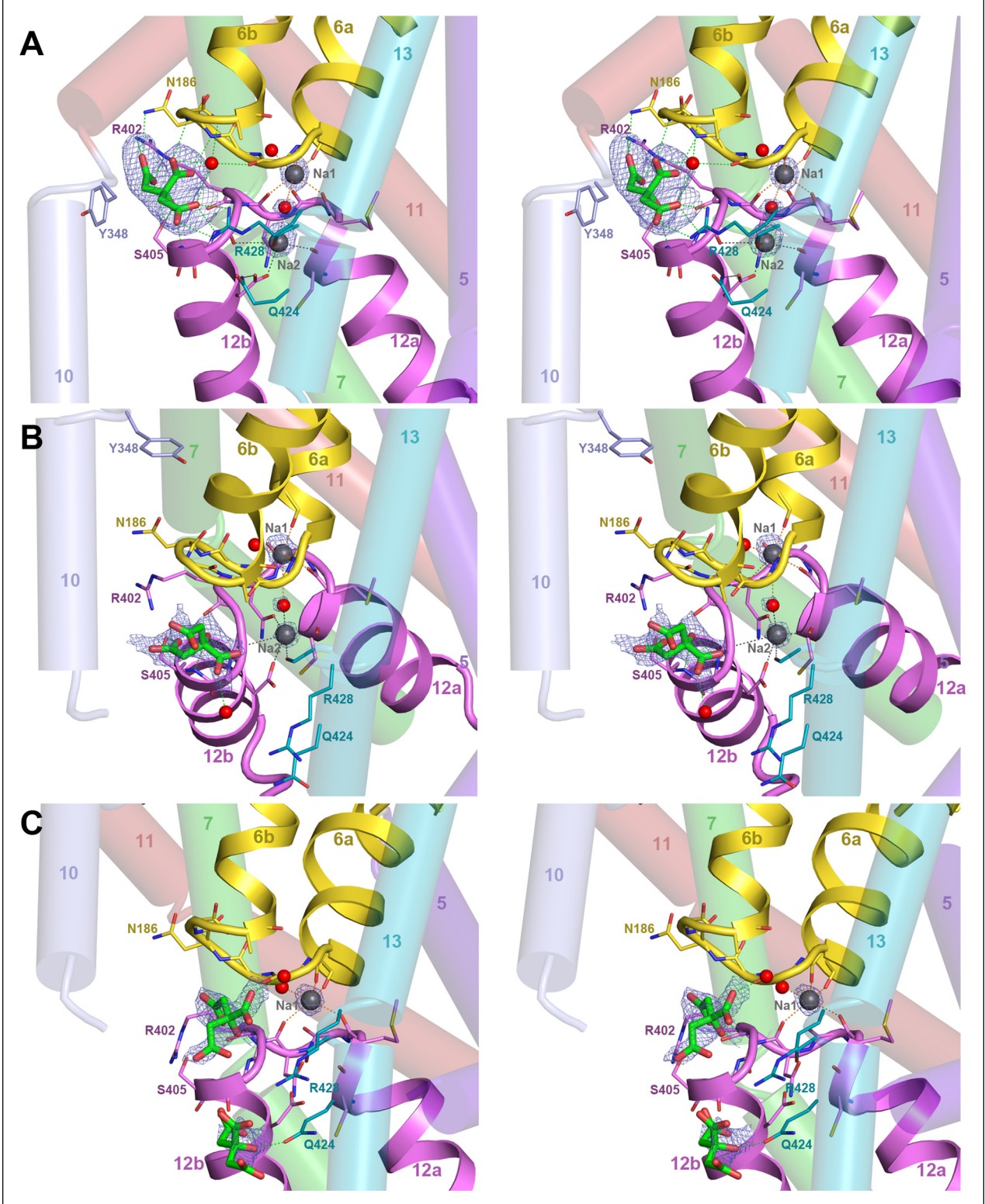

**Figure 7.** Binding sites and $F_o$-$F_c$ ligand density. (**A**) Stereo view of the outward-facing substrate-binding site of protomer A with an extensively coordinated citrate molecule. (**B**) In the inward-facing binding site of protomer B the citrate is attached less strongly. In (**A**) and (**B**) the $F_o$-$F_c$ density (blue mesh) is contoured at 3σ for citrate and at 5σ for the two bound $Na^+$ ions and the water molecule between them. (**C**) In the inward-facing protomer B', the $F_o$-$F_c$ map contoured at 4σ shows an occupied Na1 site, while the Na2 site is empty. The $F_o$-$F_c$ omit map contoured at 2.5 clearly shows two citrate molecules.

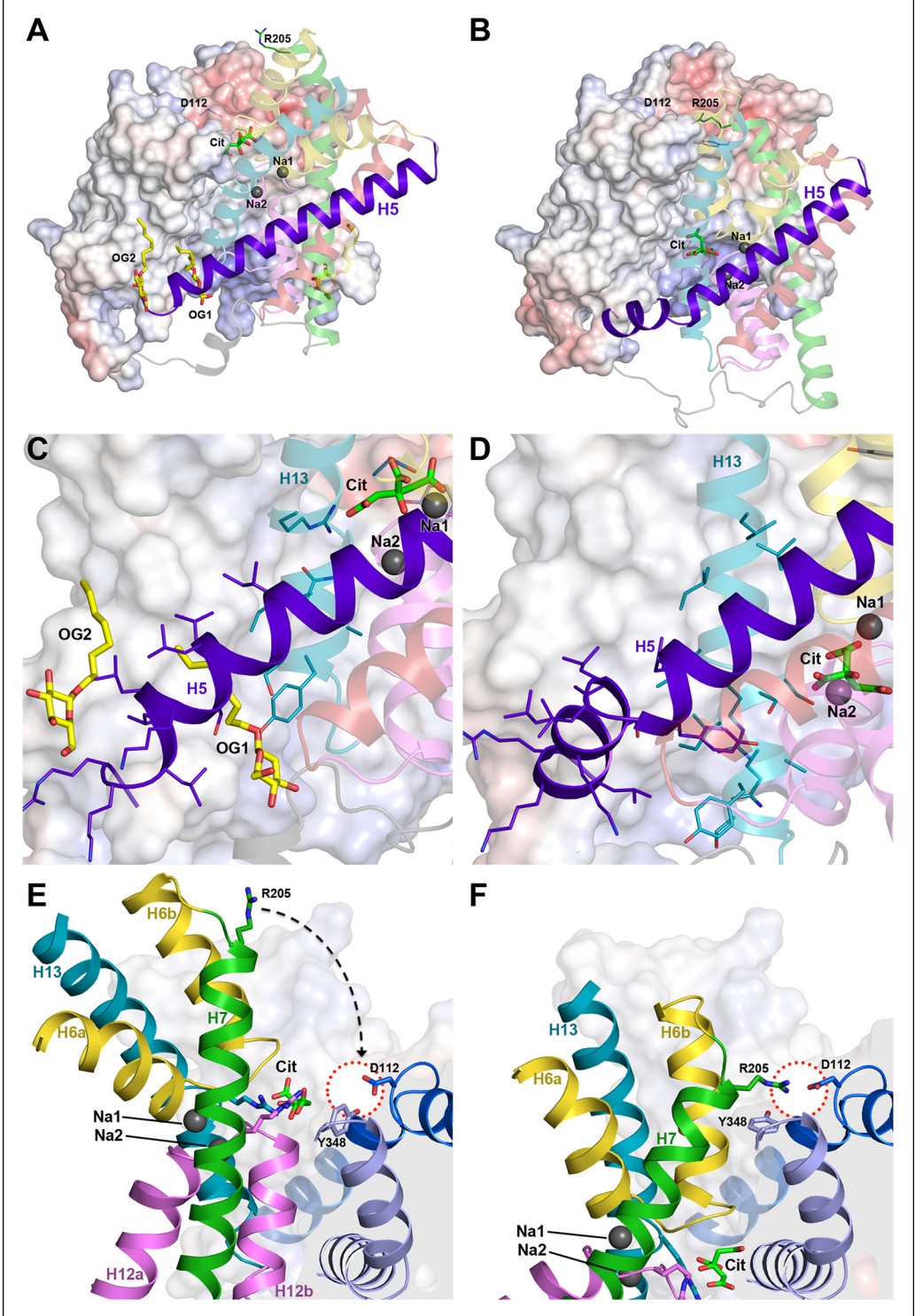

**Figure 8.** Hydrophobic interface between helix bundle and dimer contact domain. (**A, C**) In the outward-facing protomers A and A', a hydrophobic pocket between helix H5, H13 and the dimer contact domain harbors a detergent molecule that apparently replaces a membrane lipid. (**B, D**) In the inward-facing protomers B and B' H5 kinks at Gly143 and shifts towards the cytoplasm. We assume that H13 fills this hydrophobic cavity in the inward-facing state. (**E**) In the outward-facing protomers, Tyr348 coordinates the citrate by π-π-interactions. As a result of the arc-like helix bundle rotation, an ion bridge forms between Asp112 and Arg205 (H7) in the inward-facing protomers (**F**). Arg205 moves by more than 20 Å from its position in the outward-facing conformation (**E**). The sidechain of Tyr348 rotates by 90°, blocking the entrance to the substrate binding site (**F**).

a microfluidizer (M-110L, Microfluidics). Unbroken cells and cell debris were removed by centrifugation at 18,000 g for 30 min. Membranes were isolated by centrifugation at 100,000 g for 1 h and resuspended at a total protein concentration of 15 mg/ml in 20 mM Tris/HCl, 140 mM choline chloride, 250 mM sucrose, 1 mM Na-citrate, 5 mM β-ME. SeCitS was solubilized by 1:1 dilution of membranes with 20 mM Tris/HCl pH7.4, 150 mM NaCl, 3% n-decyl-β-D-maltopyranoside (DM), 1 mM Na-citrate, 5 mM β-ME. Unsolubilized material was removed by ultracentrifugation at 100,000 g for 1h. The supernatant was supplemented with 45 mM imidazole and incubated with Ni-NTA beads equilibrated with 20 mM Tris/HCl pH7.4, 300 mM NaCl, 45 mM imidazole, 1 mM Na-citrate, 0.15% DM, 5 mM β-ME for 2h at 4°C. The mixture was loaded on a column and washed with equilibration buffer to remove unspecifically bound protein. For on-column cleavage the buffer was changed to 10 mM Tris/HCl pH8.2, 150 mM NaCl, 2.5 mM CaCl$_2$, 1 mM Na-citrate, 0.15% DM. Thrombin was added to the beads to a concentration of 1 U/mg protein and incubated overnight under constant mixing. The beads were washed with exchange buffer to recover tag-free SeCitS and the protein was concentrated to 5 mg/ml (50 kDa cut-off). The concentrated protein was applied to a Superdex-200 size exclusion column equilibrated with 20 mM Tris/HCl pH8.2, 150 mM NaCl, 1 mM Na-citrate, 0.15% DM, 1 mM TCEP (Tris-(2-carboxyethyl)phosphine). Fractions containing SeCitS were pooled, concentrated as above, frozen in liquid nitrogen and stored at -80°C.

**Table 1.** Data collection and refinement statistics

| | Native SeCitS | SeMet SeCitS |
|---|---|---|
| **Data collection** | SLS PXII | |
| | | |
| Wavelength (Å) | 0.979 | 0.980 |
| Space group | P1 | P2$_1$ |
| Cell dimensions | | |
| a, b, c (Å) | 86.4, 89.9, 91.8 | 90.9, 168.8, 97.9 |
| α, β, γ (°) | 90.4, 113.8, 99.5 | 90.0, 91.0, 90.0 |
| Resolution (Å) | 47.98 – 2.5 (2.6 – 2.5) | 48.95 – 3.9 (4.0 — 3.9) |
| $R_{pim}$ | 0.052 (0.872) | 0.038 (0.539) |
| I / σI | 8.9 (1.3) | 16.8 (2.2) |
| CC* | 0.999 (0.828) | 1.000 (0.944) |
| Completeness (%) | 98.8 (98.1) | 100 (100) |
| Multiplicity | 8.2 (8.1) | 41.4 (40.9) |
| **Refinement** | | |
| Resolution (Å) | 47.98 – 2.5 (2.6 – 2.5) | |
| Unique reflections | 84765 | |
| Reflections in test set | 4193 | |
| $R_{work}$/$R_{free}$ (%) | 21.0/24.8 (33.6/36.3) | |
| CC(work)/CC(free) | 0.848/0.742 (0.796/0.773) | |
| Average B-Factor (Å$^2$) | 70 | |
| No. atoms in AU | 13270 | |
| Protein | 12916 | |
| Ligands | 285 | |
| Water | 69 | |
| r.m.s. deviations: | | |
| Bond lengths (Å) | 0.003 | |
| Bond angles (°) | 0.762 | |

Values for the highest resolution shell are shown in parentheses

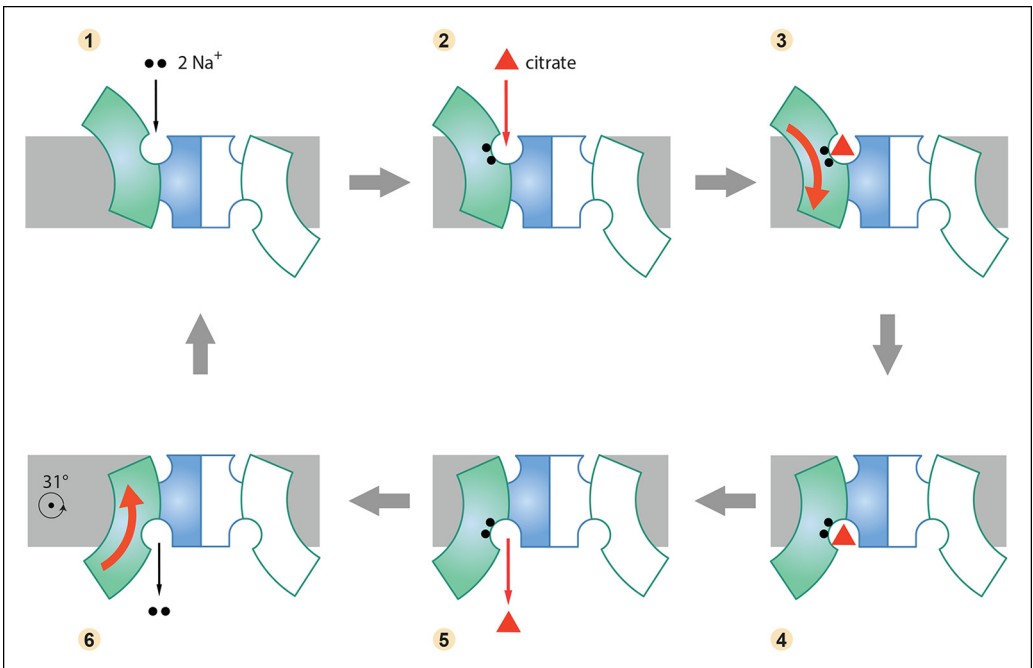

**Figure 9.** Six-step mechanism of $Na^+$-dependent citrate uptake by SeCitS. (1) Two $Na^+$ bind to the empty transporter; (2) citrate from the external medium attaches to the binding site; (3) the substrates are translocated across the membrane through a rigid-body 31° rotation of the helix bundle domain; (4) first the citrate and then (5) the $Na^+$ ions are released to the cytoplasm; (6) the unloaded protomer changes its conformation back to the outward-facing state and the cycle restarts. In the cell, the inward-directed $Na^+$ gradient drives citrate uptake, but all steps are in principle reversible. The approximate position of the rotation axis parallel to the membrane and perpendicular to the long dimer axis is indicated in (6).

Selenomethionine (SeMet)-substituted protein was expressed in a defined medium by methionine biosynthesis inhibition (*Doublie, 2007*). Expression cultures were directly inoculated with pre-cultures grown in non-inducing PA-0.5G medium (*Studier, 2005*). The main culture was grown at 37°C, induced at an $OD_{600}$ of 0.5 and harvested after 4 h. Purification of SeMet SeCitS was performed as described for the native protein.

## Crystallization and data collection

For crystallization, native SeCitS was supplemented with n-octyl-β-D-glucopyranoside (OG) to a concentration of 1%. The protein was mixed 1:1 with reservoir solution (100 mM MES pH6.5, 200 mM NaCl, 29% PEG400) and crystallized in 24-well hanging drop plates. Rhombic crystals appeared within 3 days and grew to a size of 150 μm within a week. Crystals were harvested and vitrified in liquid nitrogen using Al''s oil (*D'arcy et al., 2003*) as cryo-protectant.

SeMet-derivatized SeCitS was supplemented with 2% n-heptyl-β-D-glucopyranoside (HG) and mixed 1:1 with reservoir solution (100 mM MES pH6.5, 250 mM NaCl, 30% PEG400). Thin needle-like crystals grew to 400 μm within a week and were vitrified in liquid nitrogen directly. All datasets were collected on beamline X10SA (PXII) at the SLS (Villigen, Switzerland).

## Data processing and structure solution

All datasets were processed with XDS (*Kabsch, 1993*) and scaled with AIMLESS (*Evans, 2006*) from the CCP4 package (*Winn et al., 2011*). Resolution limits were based on I/σ(I)-values, completeness and cross correlation of half datasets (*Karplus and Diederichs, 2012*) in the high-resolution shells. PHENIX (*Adams et al., 2010*) and Coot (*Emsley et al., 2010*) were used for refinement and model building, respectively. Experimental phases were obtained by single-wavelength anomalous dispersion (SAD) from SeMet-derivatized SeCitS. Initial SeMet positions were determined by SHELXD (*Schneider and Sheldrick, 2002*) through the HKL2MAP (*Pape and Schneider, 2004*) interface and

fed into Crank2 (*Skubak and Pannu, 2013*) for substructure refinement, phasing with Refmac (*Murshudov et al., 1997*), hand determination, initial density modification with Parrot (*Zhang et al., 1997*) and model building using Buccaneer (*Cowtan, 2006*). An initial backbone model of SeCitS was created for phasing of the native high-resolution data by molecular replacement with PHASER (*McCoy et al., 2007*). Model building was performed by PHENIX autobuild (*Terwilliger et al., 2008*), followed by cycles of manual model building and refinement. Superimpositions were performed with GESAMT (*Winn et al., 2011*). Figures were drawn and rmsd values were calculated with PyMOL (*DeLano and Lam, 2005*). Electrostatic surfaces were calculated with PDB2PQR (*Dolinsky et al., 2004*) and APBS (*Baker et al., 2001*).

## Reconstitution into liposomes

*E. coli* polar lipids in chloroform (Avanti Polar Lipids) were dried under nitrogen and resuspended in reconstitution buffer (20 mM Tris/BisTris/Acetate pH 4-–8, 50 mM choline chloride), supplemented with 15 mM β-ME. Unilamellar ~400 nm vesicles were prepared using polycarbonate filters in an extruder (Avestin). Preformed liposomes were diluted to 5 mg/ml in reconstitution buffer and destabilized by addition of 1% OG. SeCitS was added at a lipid-to-protein ratio of 50 and incubated for 1 h. The protein/lipid mixture was filled into dialysis bags (14 kDa cutoff) and dialyzed against detergent-free reconstitution buffer overnight. Biobeads were added to the dialysis buffer to facilitate complete detergent removal. The proteoliposomes were centrifuged for 25 min at 300,000 g and resuspended in fresh reconstitution buffer.

## Transport measurements

Transport activity was measured with $[1,5]^{14}C$-citrate or $1,4(2,3)$-$^{14}C$-malate as a reporter molecule. Measurements were started by dilution of 2 µl freshly prepared proteoliposomes into 200 µl reaction buffer (20 mM Tris/BisTris/acetate pH 5–8, 50 mM NaCl, 5 µM $[1,5]^{14}C$-citrate or 43 µM $1,4(2,3)$-$^{14}C$-malate). Within the linear range of uptake, 200 µl samples were transferred on 0.2 µm nitrocellulose filters that were subsequently washed with 3 ml of reaction buffer. Filters were transferred into counting tubes and filled with 4 ml liquid scintillation cocktail (Rotiszint) before evaluation. All measurements were performed in triplicates. In all experiments initial rates within the linear range of uptake were recorded over a total of 4 time points.

Kinetic measurements were performed at pH6 by varying the concentration of one substrate while keeping the other constant. Ion specificity of SeCitS was determined by changing the co-substrate to LiCl, KCl or choline chloride, which is not transported. Specificity for citrate was established with a competition assay. Potential substrates were added to the reaction buffer at a concentration of 5 mM (1000x excess) to compete with $^{14}C$-citrate uptake. The effect of $\Delta pH$ on the transport activity was measured by changing the pH of the reaction buffer while keeping the inside pH constant.

## Acknowledgments

We thank Sabine Häder and Heidi Betz for technical assistance, Klaas Martinus Pos, Gerhard Hummer and Christine Ziegler for discussion, and Pavol Skubak for the Crank2 software. Crystals were screened at beamlines id23.1 and id29 of the European Synchrotron Radiation Facility (ESRF Grenoble) and data were collected at the Max Planck beamline PXII of the Swiss Light Source (SLS). This work was funded by the Max Planck Society; the Frankfurt International Max Planck Research School; and SFB 807 "Transport and communication across biological membranes".

## Additional information

### Competing interests

WK: Reviewing editor, *eLife.* The other authors declare that no competing interests exist.

## Funding

| Funder | Grant reference number | Author |
| --- | --- | --- |
| Max-Planck-Gesellschaft | Dept Struct Biol | David Wöhlert<br>Maria J Grötzinger<br>Werner Kühlbrandt<br>Özkan Yildiz |
| Deutsche Forschungsgemeinschaft | SFB807 | David Wöhlert<br>Maria J Grötzinger<br>Werner Kühlbrandt |

The funders had no role in study design, data collection and interpretation, or the decision to submit the work for publication.

## Author contributions

DW, Acquisition of data, Analysis and interpretation of data, Drafting or revising the article; MJG, Acquisition of data, Analysis and interpretation of data; WK, Conception and design, Analysis and interpretation of data, Drafting or revising the article; ÖY, Conception and design, Acquisition of data, Analysis and interpretation of data, Drafting or revising the article

## Author ORCIDs

Özkan Yildiz, http://orcid.org/0000-0003-3659-2805

# Additional files

## Major datasets

The following datasets were generated:

| Author(s) | Year | Dataset title | Dataset ID and/or URL | Database, license, and accessibility information |
| --- | --- | --- | --- | --- |
| David Wöhlert, Maria J Grötzinger, Werner Kühlbrandt, Özkan Yildiz | 2015 | Crystal structure of the sodium-dependent citrate symporter SeCitS form Salmonella enterica | http://www.rcsb.org/pdb/search/structid-Search.do?structureId=5a1s | Publicly available at the RCSB Protein Data Bank (Accession no: 5a1s). |

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
