## [Decision Letter]

Thank you for submitting your work entitled "Complete mechanism of Na^+^ -dependent citrate transport from the structure of an asymmetrical CitS dimer" for peer review at *eLife*. Your submission has been favorably evaluated by John Kuriyan (Senior Editor) and three reviewers.

This paper has received a very positive response, and all three reviewers are in agreement that this paper presents important structural results that are suitable for publication in *eLife*. The reviewers, in their individual reviews and in the online discussion, had many suggestions as to how to improve the manuscript and to properly balance the structural work with the functional studies. Rather than attempt to condense this discussion, the editor has provided the reviews in full with many of the points raised in reviewer discussion embedded within. The authors should use their own judgment in how best to respond to these comments in preparing a revised manuscript. Note that no new experimental work is called for.

Reviews and discussion comments:

*Reviewer 1:* The manuscript by Wöhlert et al. presents X-ray crystal structures at high resolution for a secondary transporter from *Salmonella enterica*. The crystallographic unit reveals four distinct protomers arranged into two dimers, with two distinct conformational states and several different substrate binding states. The structures provide extensive insights into the details of the binding sites, and the conformational change required to transition from outward-facing substrate-bound to inward-facing substrate-bound states for this transporter. Several biochemical measurements are also included using proteoliposomes under various conditions to demonstrate that the transporter can transport citrate in the presence of sodium, ruling out other di- and tri-carboxylic substrates, and testing the effect of pH, from which they conclude that the transporter does not translocate H^+^. Their measurements also indicate that malate and isocitrate are competitive inhibitors. The manuscript is clearly written and the structure analyzed in detail.

My main concerns with the manuscript relate to overinterpretation of the structural data (particularly with respect to the substrate binding/release order, and role of lipids), which in light of the extensive findings, does not seem to be necessary. In this regard, the title and Abstract should be toned down, as I do not think that the structures are sufficient to explain the entire mechanism, as there are no structures of the apo states. I also would welcome some discussion of the similarities and differences between other folds in the literature, especially VcINDY and GltPh. Finally, some issues need to be clarified with respect to the substrates being transported, particularly the species of citrate. My concerns are laid out by topic below.

1) Sodium stoichiometry: the authors state that a Hill coefficient of 1.89 suggests that coupling requires two Na^+^ (Results and Discussion, first paragraph). This is not correct, a Hill coefficient > 1 means that more than one Na^+^ ion is involved; higher stoichiometries of 3 or more cannot be ruled out from the Hill coefficient. Although the structures show evidence for two sodium ions, neither the Hill coefficient, nor the 2.5 Angstrom structure, can rule out co-transport of a third (or fourth) Na^+^ ion. Since related proteins couple to three sodium ions, this is an important point.

2) Role of protons and citrate protonation state (Results and Discussion, first paragraph onwards): I found the arguments relating to the effect of protons and the citrate protonation state needed to be better argued and supported. First, "van der Rest" et al. 1992 is cited as evidence that CitS transports HCit^2-^. There seem to be two problems with this citation: the van der Rest paper studies *Klebsiella pneumoniae* CitS, not *Salmonella enteric*, and it is entirely possible that they transport different species. At the least, the reasons to make that assumption should be explored and described, including mentioning the p*K*_a_'s of the different protonation states of citrate. In addition, van der Rest et al. concluded that the stoichiometry is one sodium and two protons, whereas the current work concludes that sodium is cotransported but not protons. The roles of the different species are interconnected, so these two different scenarios could lead to different conclusions about the citrate protonation state in order that the process is electroneutral. It is therefore important that the arguments or uncertainties relating to the transport stoichiometry and species be laid out more clearly.

On a similar note, regarding the pH dependence, the authors invoke a role for sodium at low pH; first, I'm not sure I understand the argument. Moreover, a recent work by Mulligan et al. (PMID: 24821967) follows the pH-dependence of succinate transport by a related protein VcINDY, and shows that a role for sodium need not be invoked to explain complex pH-dependence profiles, at least for succinate.

3) Detergent molecule in the interface between H5/H13 and dimerization domain (legend for Figure 5; Results and Discussion, seventh paragraph; legend for Figure 7; and legend for Video 2): The fact that a detergent is present in the outward-facing conformation does not indicate that a membrane lipid would "replace it" and lead to "participation of a membrane lipid in the transport mechanism". The authors may have been lucky that OG binds there, as it may be stabilizing the outward-facing state in the crystal, but OG is not the same as a physiologically relevant phospholipid. Indeed, in a different secondary transporter called LeuT, an OG detergent molecule binds to the extracellular cavity and stabilizes the outward-facing conformation (Nissen and coworkers). No-one would argue that this site is required for a lipid interaction during transport. These references (including to H13 "dislodging" the detergent molecule") should be removed. Any speculation with respect to the role of this detergent should be clearly labeled as speculation.

4) Ligand binding site: please replace the word "tightly" with "closely" when describing the ligand interactions. Close proximity in a crystal structure, while suggestive, does not equate to strength. Similarly, the reference to Tyr348 being "held in place" by Asp112, is speculative. Biochemical evidence, and molecular simulations would be useful to support such assertions, but even then, it is difficult to state conclusively that such interactions are maintained outside of the crystal lattice, and/or are important for function. Indeed, D112 is not conserved across other homologues shown in Figure 3—figure supplement 2, and nor is R205 or Y348. Moreover, for a "strong" interaction it is surprising to see a difference in the orientation of one of the citrate carboxylates groups, as seen when comparing protomer A with protomer A' in Figure 4.

The density assigned to citrate in the inward-facing protomer B, and the one assigned to the second citrate in the inward-facing protomer B', is significantly less well resolved, and it is not clear that they fit citrate as well as in the other site. The authors should discuss the possibility that some other (crystallization buffer) molecule occupies those regions.

5) The presence of a state with one sodium ion and one citrate, but one of the sodium sites empty, leads to over-interpretation of the substrate release steps (Results and Discussion, ninth paragraph). Since the citrate is not completely removed from the pathway and presumably blocks it, it is not clear how the missing sodium is supposed to have "released". This issue becomes further contradictory when visualizing the movie (s), which are shown as one citrate and one sodium coming off together (see below), and then when looking at the schematic of the transport cycle (Figure 8), when the substrate, and then the sodiums, come off one after another. I strongly recommend that these descriptions of the steps of unbinding be mentioned only in passing and clearly described as speculation. Such detail is not necessary as these are already, in many other ways, very informative structures.

6) The authors do not mention what the crystal packing is like. Are any of the contacts potentially relevant?

7) The videos should show only the states that are known. At most, it is reasonable to show the morphed transition between the bound forms. The binding steps are too speculative. First, the sodiums bind essentially to the same state of the protein as the substrate-bound form (or one in which some side chains have been selected to move out of the way). There is no known structure of the sodium-free or substrate free forms of SeCitS, and evidence from other proteins shows that the protein backbone changes when those substrates are missing; therefore it is misleading to suggest that the structure remains the same. Similarly in the inward-facing states, it is implied by the movie that one sodium and citrate come off before the second ion; however none of those intermediates are known; in fact, they contradict the occupancy shown in protomers B and B'.

*Reviewer 2:*Yildiz and colleagues describes a remarkable structure of a bacterial citrate transporter, CitS. The dimeric transporter is captured in the crystal in an asymmetric state such that one domain occupies an outward position and the other is inward. Thus, this single structure reveals the key aspects of CitS transport mechanism. Specifically, the structure shows that CitS belongs to a growing class of secondary active transporters that function by an elevator-like mechanism. In these transporters, one domain carrying the bound substrate and coupled ions moves across the bilayer along a proteinous track provided by another, stationary domain, which is involved in subunit association. Despite some concerns, I feel that overall the work is technically sound and most conclusions are well founded. I do not have major concerns that would require extensive rebuttal. However, I feel that the manuscript could be substantially improved through editorial changes, and I have a number of questions and suggestions. These are only minor comments. Overall, these fall into three categories: (A) concerns regarding the presentation and discussion of the functional data, including certain vagueness of the methods; (B) information content of the structural figures and corresponding structural discussions, which I think can be significantly strengthened; (C) putting results into overall perspective, which I feel could also be improved.

A) Concerns regarding functional data:

1) In all functional figures, it is unclear what does "Relative transport activity" constitute. Is it v_max_, initial rate at a sub-saturating substrate concentration, a measurement at a single time point? If the latter, it would be great to see how does typical time dependence look like? Similarly, when K_m_ values were determined, was the initial rate measured for each concentration of the substrate/ion? The methods are not very clear on these aspects.

Related comments raised during the consultation between the reviewers:

The Materials and methods describes the "Transport measurements". There, they report that they counted total radioactivity of captured C14, measuring how much radioactive citrate was imported into liposomes. Perhaps move this up to main text to avoid confusion.

My comment regarding the ambiguity of the term "activity" stems from the following consideration. Ideally, when comparing activity under different conditions (substrate concentration, pH etc.) one would want to plot the initial rates. This is why I would have liked to see how does the time dependence of the uptake look like: for how long the uptake is linear and when does it plateau. Such plot together with the note on which time point was used to measure the activity in the remaining experiments would have provided a clearer more rigorous description of the data.

2) In Figure 2, I am surprised by the substantial variability of background uptake in the absence of the protein. For example, it is particularly high in panel C. What is the origin of this variability?

Related comments raised during the consultation between the reviewers:

Figure 1 show uptake and its inhibition, Figure 1 shows how the substrate drives the uptake.

According to one of the reviewers, in Figure 1, it seems that the authors measured how much C14 was able to enter liposomes, when different buffers were on the outside. In (D), they show that C14-citrate can always enter, as long as Na and Citrate are outside, even in the presence of additional Li^+^ or K^+^ . In (C) they show that C14-citrate can only enter if Na^+^ is present on the outside, but not if Na^+^ is absent and K^+^ or Li^+^ are present instead. The referees agree that here the high C14 uptake in the absence of protein is strange and needs further explanation. Did the authors temporarily break the liposomes when adding salts, due to osmotic imbalances, and that is how C14 could enter the liposomes?

Very high background in Figure 1 (uptake by naked liposomes is ~40% of that in the presence of Na^+^ gradient) is worrying, but not crucial.

3) It would be helpful if the authors commented whether the turnover rate of 1.2 per minute is slower than expected. Is such rate physiologically relevant? Could it be that citrate is not the physiological substrate?

4) In the legend for Figure 2, it would be helpful if the authors included the concentration of Na^+^ ions used in panel A and citrate in panel B.

5) In the first paragraph of the Results and Discussion, "hill" should be capitalized. Hill coefficient above 1 suggests that at least 2 Na^+^ ions are required for substrate binding, but does not exclude the possibility that more than 2 Na^+^ ions are involved. Hence the sentence should read "coupled to at least two Na^+^ ions".

6) Was background (uptake in the absence of Na^+^ gradient) subtracted in Figure 2–figure supplement 1 and supplement 2? If not, what fraction of the observed uptake at pH 5 and 8 or in the presence of internal monovalent cations is due to background? How does background uptake depend on external pH in Figure 2–figure supplement 2A? Can that dependence explain increased uptake in the lower external pH buffer?

7) I find the observation that the symmetric low pH inhibits transport while asymmetric external low pH stimulates uptake very interesting. However, the authors' explanation of this phenomenon seems unsatisfying. They suggest that inhibition in symmetric low pH could be due to limiting Na^+^ binding. If so, it should also manifest in asymmetric conditions. In contrast, the suggestion that low pH increases concentration of transportable doubly protonated citrate species explaining activation in asymmetric conditions seems reasonable. Could it be that low internal pH inhibits the transporter because some protonatable protein groups with a near neutral p*K*_a_ need to be deprotonated to allow the return into the outward facing state?

Related comments raised during the consultation between the reviewers:

How are the authors sure that the liposomes stayed intact under all experiments?

8) The inhibition by internal monovalent cations is striking. What is a possible explanation of the observed inhibition by all monovalent cations?

Related comments raised during the consultation between the reviewers:

How are the authors sure that the liposomes stayed intact under all experiments?

One of the referee’s concerns is that the authors over-interpret the steps of substrate release and binding, given the lack of an apo structure. Moreover, their assertions do not seem to be necessary given that the overall conformational change and the binding sites are clearly shown and are already informative.

Another reviewer considered that redoing all functional experiments in a time-dependent manner might be too much. Just clarifying how the experiments were done (showing that the measurements were taken during linear phase of uptake) is sufficient. The authors’ interpretation needs to be phrased in more speculative terms. Similarly for the mechanistic discussion based on the structure; it has to be clearly stated that the proposed mechanism is speculative.

B) Concerns regarding structural figures and discussion:

9) It would be helpful if authors referenced Figure 3—figure supplement 2 early, when describing the overall structure, to make it easier for the reader to follow the discussion.

10) In the fifth paragraph of the Results and Discussion the authors say that citrate is coordinated by "backbone carbonyls of both hairpins". They should probably add that backbone amino groups also contribute to coordination of the substrate.

11) Panels C–E in Figure 4 seem to be redundant with Figure 6. On the other hand, I feel that it would be very helpful to show the close-ups of individual substrate and Na ^+^ binding sites with all coordinating residues and how they change in protomer A, B and B'. Some superpositions might be helpful to show subtle changes. Also, is it clear from how citrate is coordinated which carboxyls are protonated and which is not?

12) Some of the side chains involved in substrate/ion coordination are not labeled in Figure 4 and/or Figure 6, making it difficult to follow the discussion in the main text.

13) Figure 5 is underused. It is only referred to briefly and does not seem to contain information pertinent to the discussion.

14) I am not sure what is meant by "pi stacking with a citrate carboxyl". I might be wrong, but I thought that "stacking" generally referred to stacking interactions between aromatic rings. Also from the figure it does not look like the aromatic ring of Y348 interacts with citrate carboxyl. Instead, it looks like there is an interaction between the ring and the carbons of the substrate.

15) For the domain movement in GltPh (Results and Discussion, last paragraph), the authors should cite Reyes et al., Nature 2009 paper.

16) It seems to me that the authors could further capitalize on their structures to explain mechanistic features. For example, the bundle domain has to be able to rotate when it is fully bound but not when it is bound to only Na^+^ ion. Does it transpire from the structures why fully bound bundle can move (protomers A) but only Na^+^ -bound (protomer B') cannot? It would be also interesting to see the surface of the bundle domain that faces the scaffold domain colored by local electrostatics. Is it hydrophobic or is it polar? Are there exposed charged groups that have to move across the interface? Does the property of the surface change in citrate bound (A) and citrate free (B') structures?

Related comments raised during the consultation between the reviewers:

This might be beyond the scope of this paper. This manuscript would be a good base for molecular dynamics simulations and follow-up interpretations.

C) Discussion and perspective:

17) Discussion of inverted repeats in relation to the mechanism could be helpful here. Also comparison with other "elevator" proteins could perhaps be improved and maybe accompanied by a figure that would emphasize common and/or distinct features.

18) I did not understand the kinetic argument in the eleventh paragraph of the Results and Discussion. The fact that KpCitS is fast and SeCitS is slow in proteoliposomes does not seem to necessarily imply that rotation movement of the substrate-loaded domains is not rate limiting. That rate could, in principle, be protein-dependent. In contrast to the remarks regarding GltPh, in that protein the movements of the loaded domain do appear to be rate limiting based on Akyuz et al. (Nature, 2013 and Nature 2015). I also do not understand the sentence "Part of the energy to drive this slow step may be stored in the kinked helix H5, which straightens like a spring in the transition to the outward-facing state." The helix would only function as a "spring" if its kinked conformation were energetically unfavorable. But are there any reasons to suggest that it is? Also the following sentence seems vague.

*Reviewer 3:* Wöhlert et al. present the structure of the Citrate-Sodium Symporter CitS from *Salmolenna enterica*. They have determined the crystallographic structure of SeCitS to 2.5Å resolution, with the asymmetric crystal unit cell containing two homodimers in different conformations, i.e., four monomers. This allowed them to observe the monomer in three different conformations. The homo-dimeric CitS thereby was found to have drastically different conformations for the two monomers, one in the outwards open state and one in the inwards open state. Slightly different third conformations were found for the inwards open states, giving three conformations in total. This allowed the authors to draw conclusions about the mechanism of symport, which is in detail described.

The manuscript is excellently written and convincing. It includes extensive functional activity tests, and is well agreeing with earlier work by 2D crystals on a related CitS. The mechanism of translocation and conformational changes described are new, however.

---

## [Author Response]

Reviewer 1:

*My main concerns with the manuscript relate to overinterpretation of the structural data (particularly with respect to the substrate binding/release order, and role of lipids), which in light of the extensive findings, does not seem to be necessary. In this regard, the title and Abstract should be toned down, as I do not think that the structures are sufficient to explain the entire mechanism, as there are no structures of the apo states. I also would welcome some discussion of the similarities and differences between other folds in the literature, especially VcINDY and GltPh. Finally, some issues need to be clarified with respect to the substrates being transported, particularly the species of citrate. My concerns are laid out by topic below.*

We removed "Complete" from the title, which now reads "Mechanism of Na^+^ -dependent citrate transport from the structure of an asymmetrical CitS dimer".

*1) Sodium stoichiometry: the authors state that a Hill coefficient of 1.89 suggests that coupling requires two Na ^+^ (Results and Discussion, first paragraph). This is not correct, a Hill coefficient > 1 means that more than one Na^+^ ion is involved; higher stoichiometries of 3 or more cannot be ruled out from the Hill coefficient. Although the structures show evidence for two sodium ions, neither the Hill coefficient, nor the 2.5 Angstrom structure, can rule out co-transport of a third (or fourth) Na^+^ ion. Since related proteins couple to three sodium ions, this is an important point.* We changed the statement "… citrate transport is obligatory coupled to two Na^+^ ions" to "… citrate transport is coupled to at least two Na^+^ ions".

*2) Role of protons and citrate protonation state (Results and Discussion, first paragraph onwards):I found the arguments relating to the effect of protons and the citrate protonation state needed to be better argued and supported. First, "van der Rest" et al. 1992 is cited as evidence that CitS transports HCit^2-^. There seem to be two problems with this citation: the van der Rest paper studies* Klebsiella pneumoniae *CitS, not* Salmonella enteric*, and it is entirely possible that they transport different species. At the least, the reasons to make that assumption should be explored and described, including mentioning the p*K*_a_'s of the different protonation states of citrate.*

Although it is in principle not impossible that KpCitS and SeCitS transport differently charged citrate ions, we consider this extremely unlikely. One reason is the high level of sequence identity of 92% between SeCitS and KpCitS. Identical residues include in particular those that form the substrate-binding pocket. Such a high level of sequence identity indicates that the structures of the proteins, and hence their mechanisms, must be essentially the same.

As requested by referee #2 (comment 9), the sequence alignment (Figure 3—figure supplement 2 in the original manuscript) is now Figure 1 in the revised manuscript. All subsequent figure numbers have gone up by 1. We have added a short passage on the sequence homology and what we conclude from it (Results and Discussion, first paragraph).

*In addition, van der Rest et al. concluded that the stoichiometry is one sodium and two protons, whereas the current work concludes that sodium is cotransported but not protons. The roles of the different species are interconnected, so these two different scenarios could lead to different conclusions about the citrate protonation state in order that the process is electroneutral. It is therefore important that the arguments or uncertainties relating to the transport stoichiometry and species be laid out more clearly.*

Initially van der Rest et al. proposed that CitS co-transports HCit^2-^ together with one Na^+^ and two H^+^ (van der Rest et al. 1992). Later the same group demonstrated that HCit^2-^ is co-transported with two Na^+^ ions (Lolkema 1994). Both Lolkema et al. and van der Rest et al. used whole-cell measurements to determine transport activity, which could have resulted in a high background from other membrane transporters. In 1996 Pos et al. verified the coupling of citrate transport to at least two sodium ions and demonstrated electroneutral transport by KpCitS, both in proteoliposomes, as we now do for SeCitS. In addition, Pos et al. demonstrated a stimulating effect of ΔpH on citrate transport, which was interpreted in the context of the antiport mechanism (OH^-^ antiport/ proton symport). All these studies agree on the fact that KpCitS and SeCitS take up divalent citrate. This is now explained briefly in of the revised manuscript.

*On a similar note, regarding the pH dependence, the authors invoke a role for sodium at low pH; first, I'm not sure I understand the argument. Moreover, a recent work by Mulligan et al. (PMID: 24821967) follows the pH-dependence of succinate transport by a related protein VcINDY, and shows that a role for sodium need not be invoked to explain complex pH-dependence profiles, at least for succinate.*

The p*K*_a_ values for citrate and succinate ions are:

Succinate: 1. pH 4.2 2. pH 5.6

Citrate: 1. pH 3.1 2. pH 4.8 3. pH 6.4

In our measurements transport drops to background level at pH 5. Using the p*K*_a_ values above we calculate a theoretical fractional composition of divalent citrate at various pH values as follows:

Cit^2-^

pH 5.5: 76.4%

pH 6.0: 68.7%

pH 6.5: 43.9%

pH 7.0: 20.1%

While the amount of potential HCit^2-^ substrate increases substantially at lower pH, the transport activity of SeCitS drops. It is therefore clear that the pH profile of SeCitS is not simply explained by the p*K*_a_ of citrate alone. Instead, we conclude that at low pH, sodium binding becomes limiting. Our conclusion is substantiated by Lolkema et al. (1994), who report a competition of sodium ions and protons at the sodium-binding site. Lolkema et al. also demonstrated that protons cannot replace sodium during transport.

A detailed discussion of this point can be found in the Results and Discussion.

3) Detergent molecule in the interface between H5/H13 and dimerization domain (legend for Figure 5; Results and Discussion, seventh paragraph; legend for Figure 7; and legend for Video 2): The fact that a detergent is present in the outward-facing conformation does not indicate that a membrane lipid would "replace it" and lead to "participation of a membrane lipid in the transport mechanism". The authors may have been lucky that OG binds there, as it may be stabilizing the outward-facing state in the crystal, but OG is not the same as a physiologically relevant phospholipid. Indeed, in a different secondary transporter called LeuT, an OG detergent molecule binds to the extracellular cavity and stabilizes the outward-facing conformation (Nissen and coworkers). No-one would argue that this site is required for a lipid interaction during transport. These references (including to H13 "dislodging" the detergent molecule") should be removed. Any speculation with respect to the role of this detergent should be clearly labeled as speculation.

We deleted ’’taking the place of a membrane lipid “(legend for Video 2). We are still convinced that this is the case, because it is hard to see why else there would be a detergent in this position. It is however difficult to prove that it does replace a lipid, and therefore we replaced "… apparently replacing a membrane lipid"

for "… that may take the place of a membrane lipid alkyl chain."

We refer the reviewer to the study by Quick et al. (2009), “Binding of an octylglucoside detergent (OG) molecule in the second substrate (S2) site of LeuT establishes an inhibitor-bound conformation”. In LeuT the OG molecule has an inhibitory effect, blocking the S2 site from the outside. In SeCitS, OG binds to an intracellular cavity, which rules out a similar effect as in LeuT. This conclusion is further supported by the fact that our structure shows both an inward- and an outward-facing protomer, which are present simultaneously under identical conditions. If the bound detergent were to stabilize the outward-facing conformation, we would expect both protomers in the dimer to be in this conformation. Nevertheless we cannot exclude that a detergent molecule in this position stabilizes the outward-facing conformation. We conclude that lipids are likely to be present at this position, especially in the context of the recent publication by Akyuz (2015), who performed MD simulations to show that lipids can stably insert between both domains of GltPh. This paper was cited in the original manuscript, now in the Results and Discussion.

*These references (including to H13 "dislodging" the detergent molecule" should be removed; any speculation with respect to the role of this detergent should be clearly labeled as speculation.*

We have modified the revised manuscript as follows (legend for Figure 8): “We assume that H13 fills this hydrophobic cavity in the inward-facing state”.

4) Ligand binding site: please replace the word "tightly" with "closely" when describing the ligand interactions. Close proximity in a crystal structure, while suggestive, does not equate to strength. Similarly, the reference to Tyr348 being "held in place" by Asp112, is speculative. Biochemical evidence, and molecular simulations would be useful to support such assertions, but even then, it is difficult to state conclusively that such interactions are maintained outside of the crystal lattice, and/or are important for function. Indeed, D112 is not conserved across other homologues shown in Figure 3—figure supplement 2, and nor is R205 or Y348.

We replaced “tightly” for “closely” and removed “held in place”.

*Moreover, for a "strong" interaction it is surprising to see a difference in the orientation of one of the citrate carboxylates groups, as seen when comparing protomer A with protomer A' in Figure 4.*

Figure 4 shows protomer A. The only figure that includes protomer A’ is 4A. The coordination of citrate in protomers A and A’ is identical. We have added a short passage in the Results and Discussion to better explain the four protomers A, A’, B, B’ and the differences and similarities between them.

*The density assigned to citrate in the inward-facing protomer B, and the one assigned to the second citrate in the inward-facing protomer B', is significantly less well resolved, and it is not clear that they fit citrate as well as in the other site. The authors should discuss the possibility that some other (crystallization buffer) molecule occupies those regions.*

The figures show omit maps that avoid model bias. It is true that the electron densities for citrate in the inward-facing protomers are less well defined than in the outward-facing state. We assume that this reflects either a lower occupancy or a higher degree of flexibility. Both explanations are reasonable, as citrate is only loosely attached to the protein. The only molecules in the crystallization buffer that might account for the observed electron density are citrate, TRIS or TCEP. Refinement against each of these indicated that only citrate explains the data well. Therefore the omit densities in Figure 6 are best described by citrate molecules.

*5) The presence of a state with one sodium ion and one citrate, but one of the sodium sites empty, leads to over-interpretation of the substrate release steps (Results and Discussion, ninth paragraph). Since the citrate is not completely removed from the pathway and presumably blocks it, it is not clear how the missing sodium is supposed to have "released". This issue becomes further contradictory when visualizing the movie (s), which are shown as one citrate and one sodium coming off together (see below), and then when looking at the schematic of the transport cycle (Figure 8), when the substrate, and then the sodiums, come off one after another. I strongly recommend that these descriptions of the steps of unbinding be mentioned only in passing and clearly described as speculation. Such detail is not necessary as these are already, in many other ways, very informative structures.*

Our aim in the manuscript was to discuss requirements for substrate release, rather than the exact physiological order in which substrates are released, which would be highly dependent on substrate concentrations. Citrate is only loosely attached to the inward-facing protomer B, while both sodium ions are still bound. It is therefore reasonable to conclude that citrate release is independent from the release of either sodium ion. We further mentioned in the original manuscript (now Results and Discussion) that in this conformation the Na2 site would be easily accessible upon a slight movement of D407. As this site is empty in protomer B’, it seems inescapable that Na2 is released before Na1. Whether citrate and Na2 are released simultaneously or sequentially is impossible to tell from the structure alone. Nevertheless the release of Na2 immediately after citrate would be plausible with our structures, as the ion may attach to one of the carboxyl groups. In order to prevent any misunderstanding we state in the legend for Video 1 that the order in which substrates are released is speculative.

Due to the location of the sodium binding sites, which are covered by the bound citrate in the outward-facing protomer, we are convinced that the sodium ions bind first in the outward conformation, while in the inward-facing conformation, Na1 has to come off last, after citrate and Na2 have left. This enabled us to come up with a basic scheme of the complete transport cycle, which is necessarily speculative. This was mentioned explicitly in the legend for Video 1.

The point of Figure 8 is to illustrate substrate translocation and binding/unbinding events schematically. We did not go into details of the release order of the ions here, because we wanted to keep this schematic as simple as possible. Figure 8 and the movies do not contradict one another, as both show the same events. Of course the movies are more detailed.

*6) The authors do not mention what the crystal packing is like. Are any of the contacts potentially relevant?*


Author response image 1.shows the two dimers in the asymmetric unit (multi-coloured) surrounded by 12 symmetry-related dimers.**DOI:**
http://dx.doi.org/10.7554/eLife.09375.018

Both the interface and the helix bundle domains are involved in crystal contacts. The backbone geometry of the two dimers in the asymmetric unit is very similar, even though the crystal contacts for each dimer are different. Therefore the impact of crystal contacts on the conformation of the helix bundle is negligible. This is why we did not discuss the crystal packing in the original manuscript. We have now added a short statement to say that the observed conformational differences are not due to crystal packing (Results and Discussion).

*7) The videos should show only the states that are known. At most, it is reasonable to show the morphed transition between the bound forms. The binding steps are too speculative. First, the sodiums bind essentially to the same state of the protein as the substrate-bound form (or one in which some side chains have been selected to move out of the way). There is no known structure of the sodium-free or substrate free forms of SeCitS, and evidence from other proteins shows that the protein backbone changes when those substrates are missing; therefore it is misleading to suggest that the structure remains the same. Similarly in the inward-facing states, it is implied by the movie that one sodium and citrate come off before the second ion; however none of those intermediates are known; in fact, they contradict the occupancy shown in protomers B and B'.*

We do not share the reviewer’s point of view. The quality of our results entitles us to some mild speculation, as is generally accepted in the best structural biology papers. We stated that Video 1 is a schematic, and thus necessarily simplified representation of the transport cycle. To avoid any misunderstanding we have now included the same note in the legend of Video 2. To show substrate binding and release in a movie will be useful information for most readers at this stage, since otherwise our description would be limited to text information or Figure 8. We agree that structures of the apo-state are desirable, but it is not reasonable to imply, as the reviewer does, that this state would look completely different from the three conformations in hand, which is more than most transporters can muster.

The only protein of a similar fold with evidence for backbone changes during substrate binding is GltPh. The main substrate of SeCitS is bound between both transporter domains, while in GltPh it is occluded within the helix bundle. Therefore larger rearrangements during substrate binding in GltPh are expected, as the hairpins need to reorient to give access to this site. In SeCitS substrate binding requires only sidechain (Arg402 & Arg48) and slight main chain rearrangements in the loop regions of the helix hairpins. To us it seems evident that protomers B and B’ demonstrate the release order of sodium. In protomer B’ citrate is only loosely attached to SeCitS and hydrated, indicating that basically it has been released. On the other hand the empty Na2 site in protomer B’ indicates without doubt that Na1 is released last. Both findings together offer strong support to the order of substrate release presented in our movie.

Reviewer 2:

*Yildiz and colleagues describes a remarkable structure of a bacterial citrate transporter, CitS. The dimeric transporter is captured in the crystal in an asymmetric state such that one domain occupies an outward position and the other is inward. Thus, this single structure reveals the key aspects of CitS transport mechanism. Specifically, the structure shows that CitS belongs to a growing class of secondary active transporters that function by an elevator-like mechanism. In these transporters, one domain carrying the bound substrate and coupled ions moves across the bilayer along a proteinous track provided by another, stationary domain, which is involved in subunit association. Despite some concerns, I feel that overall the work is technically sound and most conclusions are well founded. I do not have major concerns that would require extensive rebuttal. However, I feel that the manuscript could be substantially improved through editorial changes, and I have a number of questions and suggestions. Overall, these fall into three categories: (A) concerns regarding the presentation and discussion of the functional data, including certain vagueness of the methods; (B) information content of the structural figures and corresponding structural discussions, which I think can be significantly strengthened; (C) putting results into overall perspective, which I feel could also be improved.*

We thank reviewer 2 for her or his overall constructive comments. Given that (1) we have already dealt with the criticisms of reviewer 1 in considerable detail, (2) reviewer 3 finds the manuscript excellently written and convincing, and (3) reviewer 2 has no major concerns, we feel that we can keep our response to the minor comments of this reviewer to a minimum.

*A) Concerns regarding functional data:1) In all functional figures, it is unclear what does "Relative transport activity" constitute. Is it v_max_, initial rate at a sub-saturating substrate concentration, a measurement at a single time point? If the latter, it would be great to see how does typical time dependence look like? Similarly, when K_m_ values were determined, was the initial rate measured for each concentration of the substrate/ion? The methods are not very clear on these aspects.*

In all experiments initial rates within the linear range of uptake were recorded over a total of 4 time points. This information is now included in the Materials and methods. Furthermore we noted in the figure legends which rates were set to 100%.

Related comments raised during the consultation between the reviewers:

The Materials and methods describes the "Transport measurements". There, they report that they counted total radioactivity of captured C14, measuring how much radioactive citrate was imported into liposomes. Perhaps move this up to main text to avoid confusion.

The section "Transport measurements" only describes the methods.

My comment regarding the ambiguity of the term "activity" stems from the following consideration. Ideally, when comparing activity under different conditions (substrate concentration, pH etc.) one would want to plot the initial rates. This is why I would have liked to see how does the time dependence of the uptake look like: for how long the uptake is linear and when does it plateau. Such plot together with the note on which time point was used to measure the activity in the remaining experiments would have provided a clearer more rigorous description of the data.

We added a note that initial uptake rates were measured over 4 time points during the linear range of uptake (subsection “Transport measurements”).

*2) In Figure 2, I am surprised by the substantial variability of background uptake in the absence of the protein. For example, it is particularly high in panel C. What is the origin of this variability?*

*Related comments raised during the consultation between the reviewers:*

*Figure 1 show uptake and its inhibition, Figure 1 shows how the substrate drives the uptake.*

According to one of the reviewers, in Figure 1, it seems that the authors measured how much C14 was able to enter liposomes, when different buffers were on the outside. In (D), they show that C14-citrate can always enter, as long as Na and Citrate are outside, even in the presence of additional Li^+^ or K^+^ . In (C) they show that C14-citrate can only enter if Na^+^ is present on the outside, but not if Na^+^ is absent and K^+^ or Li^+^ are present instead. The referees agree that here the high C14 uptake in the absence of protein is strange and needs further explanation. Did the authors temporarily break the liposomes when adding salts, due to osmotic imbalances, and that is how C14 could enter the liposomes?

Very high background in Figure 1 (uptake by naked liposomes is ~40% of that in the presence of Na^+^ gradient) is worrying, but not crucial.

In the revised manuscript this is Figure 2. We repeated the measurements with freshly prepared proteoliposomes, resulting in a substantially lower background.

*3) It would be helpful if the authors commented whether the turnover rate of 1.2 per minute is slower than expected. Is such rate physiologically relevant? Could it be that citrate is not the physiological substrate?*

A turnover of 1.2 per minute is not uncommon. VcINDY, which has a similar domain architecture, has a turnover of 1.6 succinate molecules per minute (Mulligan et al., 2014). For GltPh a turnover time of ~3 minutes was reported (Reyes et al., 2009) and similar turnover rates were also observed for LeuT, even though conformational changes in this transporter are smaller (Piscitelli et al., 2012). As demonstrated by Figure 2–figure supplement 2, the transport rate more than doubles when a ΔpH is applied, which may be relevant under physiological conditions. Given that the turnover rate of SeCitS is within the range of other transporters, we are convinced of its physiological relevance. Citrate is likely to be the main substrate as KpCitS is a key player of the anaerobic citrate metabolism in *Klebsiella pneumoniae* (Bott et al., 1995).

*4) In the legend for Figure 2, it would be helpful if the authors included the concentration of Na^+^ ions used in panel A and citrate in panel B.*

In the revised manuscript this is Figure 3. "25 mM Na^+^ " is added to the legend of Figure 3. "5 µM citrate" is added to the legend of Figure 3.

*5) In the first paragraph of the Results and Discussion, "hill" should be capitalized. Hill coefficient above 1 suggests that at least 2 Na ^+^ ions are required for substrate binding, but does not exclude the possibility that more than 2 Na ^+^ ions are involved. Hence the sentence should read "coupled to at least two Na ^+^ ions".*

Done.

*6) Was background (uptake in the absence of Na^+^ gradient) subtracted in Figure 2–figure supplement 1 and supplement 2? If not, what fraction of the observed uptake at pH 5 and 8 or in the presence of internal monovalent cations is due to background? How does background uptake depend on external pH in Figure 2–figure supplement 2A? Can that dependence explain increased uptake in the lower external pH buffer?*

In the revised manuscript these are Figure 3—figure supplement 1 and Figure 3—figure supplement 2.

Background of uptake was not subtracted either in Figure 3—figure supplement 1 nor supplement 2. We repeated the measurements with freshly prepared proteoliposomes (new Figure 2), resulting in a substantially lower background. Uptake in the absence of protein is clearly lower than 10% in all of the measurements now (Figure 2). We are therefore convinced that a subtraction of background has no influence on any of the results shown in the supplementary figures for Figure 3.

A systematically higher background in the experiment of Figure 3—figure supplement 2 would not increase the rate of initial uptake, which was measured over 4 time points. Instead it would increase the overall number of counts in all time points. This does not lead to a higher uptake, especially as the negative control demonstrates that background in these measurements is negligible.

*7) I find the observation that the symmetric low pH inhibits transport while asymmetric external low pH stimulates uptake very interesting. However, the authors' explanation of this phenomenon seems unsatisfying. They suggest that inhibition in symmetric low pH could be due to limiting Na^+^ binding. If so, it should also manifest in asymmetric conditions. In contrast, the suggestion that low pH increases concentration of transportable doubly protonated citrate species explaining activation in asymmetric conditions seems reasonable. Could it be that low internal pH inhibits the transporter because some protonatable protein groups with a near neutral p*K*_a_ need to be deprotonated to allow the return into the outward facing state?*

Our best guess for a protonable residue would be D407, which is present in the Na2 binding site. Protonation of this residue may leave an uncompensated charge close to domain interface, thereby preventing the translocation of the empty carrier. Lolkema et al., 1994 showed that protons can compete for sodium binding sites in CitS, but cannot replace them, adding further substance to this notion.

Related comments raised during the consultation between the reviewers:

How are the authors sure that the liposomes stayed intact under all experiments?

We kept salt concentrations inside and outside constant in order to prevent osmotic pressure, by replacing a part of choline chloride in the reconstitution buffer with sodium.

*8) The inhibition by internal monovalent cations is striking. What is a possible explanation of the observed inhibition by all monovalent cations? And how are the authors sure that the liposomes stayed intact under all experiments?*

We have shown in Figure 2 that none of the other ions are able to drive transport, which does of course not exclude that they may specifically inhibit transport on one side of the membrane. The point of showing this figure is to demonstrate that internal sodium does not increase the transport rate.

Related comments raised during the consultation between the reviewers:

*How are the authors sure that the liposomes stayed intact under all experiments?*

As explained above, we kept salt concentrations inside and outside constant in order to avoid osmotic effects, by replacing a part of choline chloride in the reconstitution buffer with sodium.

*One of the referee’s concerns is that the authors over-interpret the steps of substrate release and binding, given the lack of an apo structure. Moreover, their assertions do not seem to be necessary given that the overall conformational change and the binding sites are clearly shown and are already informative.*

*Another reviewer considered that redoing all functional experiments in a time-dependent manner might be too much. Just clarifying how the experiments were done (showing that the measurements were taken during linear phase of uptake) is sufficient. The authors’ interpretation needs to be phrased in more speculative terms. Similarly for the mechanistic discussion based on the structure; it has to be clearly stated that the proposed mechanism is speculative. B) Concerns regarding structural figures and discussion:9) It would be helpful if authors referenced Figure 3—figure supplement 2 early, when describing the overall structure, to make it easier for the reader to follow the discussion.*

Thanks, done. In the revised manuscript this is Figure 1.

*10) In the fifth paragraph of the Results and Discussion the authors say that citrate is coordinated by "backbone carbonyls of both hairpins". They should probably add that backbone amino groups also contribute to coordination of the substrate.*

This passage has been rewritten: "In the outward-facing protomers, the citrate is closely coordinated by two arginines (Arg402, Arg428), two polar sidechains (Asn186, Ser405) and the protein backbone of both hairpins (Figure 5 and Figure 7)."

*11) Panels C–E in Figure 4 seem to be redundant with Figure 6. On the other hand, I feel that it would be very helpful to show the close-ups of individual substrate and Na ^+^ binding sites with all coordinating residues and how they change in protomer A, B and B'. Some superpositions might be helpful to show subtle changes. Also, is it clear from how citrate is coordinated which carboxyls are protonated and which is not?*

The first paragraph of the Results and Discussion section discusses the protonation state of citrate. Furthermore, in the fifth and sixth paragraphs we describe the citrate coordination in great detail. We assume that the terminal carboxyls are deprotonated.

*12) Some of the side chains involved in substrate/ion coordination are not labeled in Figure 4 and/or Figure 6, making it difficult to follow the discussion in the main text.*

Introducing more labels into the figures would make them difficult to read and unclear, which would be counterproductive. We prefer not to add extra labels.

*13) Figure 5 is underused. It is only referred to briefly and does not seem to contain information pertinent to the discussion.*

This figure is now discussed in more detail in the Results and Discussion section.

*14) I am not sure what is meant by "pi stacking with a citrate carboxyl". I might be wrong, but I thought that "stacking" generally referred to stacking interactions between aromatic rings. Also from the figure it does not look like the aromatic ring of Y348 interacts with citrate carboxyl. Instead, it looks like there is an interaction between the ring and the carbons of the substrate.*

We replaced the term “pi stacking” in with “pi interactions” (fifth paragraph of the Results and Discussion).

In the citrate carboxyl group, pi electrons of the carbon and oxygens form a de-localized electron orbital. Similar to the well-known pi–pi stacking of aromatic rings, pi–pi interactions do occur between aromatic and non-aromatic systems (e.g. pi–pi interactions of Tyr and Arg sidechains (PMID:24438169, PMID:8196060). Therefore, there is no reason why the same sort of interaction between pi orbitals should not occur between the aromatic ring of Tyr348 and citrate. Please refer to Author response image 2, which shows electron density between the two pi systems, indicating a binding interaction.

Author response image 2.**DOI:**
http://dx.doi.org/10.7554/eLife.09375.019

*15) For the domain movement in GltPh (Results and Discussion, last paragraph), the authors should cite Reyes et al., Nature 2009 paper.*

Done.

16) It seems to me that the authors could further capitalize on their structures to explain mechanistic features. For example, the bundle domain has to be able to rotate when it is fully bound but not when it is bound to only Na^+^ ion. Does it transpire from the structures why fully bound bundle can move (protomers A) but only Na^+^ -bound (protomer B') cannot? It would be also interesting to see the surface of the bundle domain that faces the scaffold domain colored by local electrostatics. Is it hydrophobic or is it polar? Are there exposed charged groups that have to move across the interface? Does the property of the surface change in citrate bound (A) and citrate free (B') structures?

Related comments raised during the consultation between the reviewers:

This might be beyond the scope of this paper. This manuscript would be a good base for molecular dynamics simulations and follow-up interpretations.

Charges in a helix bundle that is only partially loaded with substrate would be uncompensated, which would likely inhibit the substrate translocation movement. Any further insight into this would require a structure of the apo state.

*C) Discussion and perspective:17) Discussion of inverted repeats in relation to the mechanism could be helpful here. Also comparison with other "elevator" proteins could perhaps be improved and maybe accompanied by a figure that would emphasize common and/or distinct features.*

We agree that these points are interesting to discuss. However we are concerned that a detailed discussion of these topics within this manuscript be more of a distraction rather than being informative, given the wealth of data that would need to be discussed.

*18) I did not understand the kinetic argument in the eleventh paragraph of the Results and Discussion. The fact that KpCitS is fast and SeCitS is slow in proteoliposomes does not seem to necessarily imply that rotation movement of the substrate-loaded domains is not rate limiting. That rate could, in principle, be protein-dependent. In contrast to the remarks regarding GltPh, in that protein the movements of the loaded domain do appear to be rate limiting based on Akyuz et al. (Nature, 2013 and Nature 2015). I also do not understand the sentence "Part of the energy to drive this slow step may be stored in the kinked helix H5, which straightens like a spring in the transition to the outward-facing state." The helix would only function as a "spring" if its kinked conformation were energetically unfavorable. But are there any reasons to suggest that it is? Also the following sentence seems vague.*

In the Results and Discussion section, we state that citrate uptake of SeCitS is substantially slower than citrate exchange in KpCitS. In uptake measurements the transporter completes the complete transport cycle, while in exchange measurements the sodium-bound transporter shuttles citrate from one side to the other. A discrepancy between uptake and exchange rates was also reported in Pos et al., 1996 for KpCitS alone, even though an exact uptake rate was not reported.

The sequence identity between KpCitS and SeCitS of 92.2%, indicating that the two structures, and therefore the transport mechanisms are essentially identical. Comparison of the uptake rate of SeCitS with the exchange rate of KpCitS implies that the movement of the substrate-loaded transporter does not limit the transport rate. We removed the spring statement and the subsequent sentence in the revised manuscript.